# Solving High Frequency and Multi-Scale PDEs with Gaussian Processes

**Shikai Fang**[*], **Madison Cooley**[*], **Da Long**[*], **Shibo Li, Robert M. Kirby, Shandian Zhe**[†]
University of Utah, Salt Lake City, UT 84112, USA
{shikai,mcooley,dl932,shibo,kirby,zhe}@cs.utah.edu

## Abstract

Machine learning based solvers have garnered much attention in physical simulation and scientific computing, with a prominent example, physics-informed neural networks (PINNs). However, PINNs often struggle to solve high-frequency and multi-scale PDEs, which can be due to the spectral bias during neural network training. To address this problem, we resort to the Gaussian process (GP) framework. To flexibly capture the dominant frequencies, we model the power spectrum of the PDE solution with a student $t$ mixture or Gaussian mixture. We apply inverse Fourier transform to obtain the covariance function (by Wiener-Khinchin theorem). The covariance derived from the Gaussian mixture spectrum corresponds to the known spectral mixture kernel. Next, we estimate the mixture weights in the log domain, which we show is equivalent to placing a Jeffreys prior. It automatically induces sparsity, prunes excessive frequencies, and adjusts the remaining toward the ground truth. Third, to enable efficient and scalable computation on massive collocation points, which are critical to capture high frequencies, we place the collocation points on a grid, and multiply our covariance function at each input dimension. We use the GP conditional mean to predict the solution and its derivatives so as to fit the boundary condition and the equation itself. As a result, we can derive a Kronecker product structure in the covariance matrix. We use Kronecker product properties and multilinear algebra to promote computational efficiency and scalability, without low-rank approximations. We show the advantage of our method in systematic experiments. The code is released at https://github.com/xuangu-fang/Gaussian-Process-Slover-for-High-Freq-PDE.

## 1 Introduction

Scientific and engineering problems often demand we solve a set of partial differential equations (PDEs). Recently, machine learning (ML) solvers have attracted much attention. Compared to traditional numerical methods, ML solvers do not require complex mesh designs and sophisticated numerical tricks, are simple to implement, and can solve inverse problems efficiently and conveniently. The most popular ML solver is the physics-informed neural network (PINN) (Raissi et al., 2019). Consider a PDE of the following general form,

$$\mathcal{F}[u](\mathbf{x}) = f(\mathbf{x}) \ (\mathbf{x} \in \Omega), \quad u(\mathbf{x}) = g(\mathbf{x}) \ (\mathbf{x} \in \partial\Omega), \tag{1}$$

where $\mathcal{F}$ is the differential operator, $\Omega$ is the domain, and $\partial\Omega$ is the boundary of the domain. To solve the PDE, the PINN uses a deep neural network (NN) $\widehat{u}_{\boldsymbol{\theta}}(\mathbf{x})$ to model the solution $u$. It samples $N_c$ collocation points $\{\mathbf{x}_c^j\}_{j=1}^{N_c}$ from $\Omega$ and $N_b$ points $\{\mathbf{x}_b^j\}_{j=1}^{N_b}$ from $\partial\Omega$, and minimizes a loss,

$$\boldsymbol{\theta}^* = \operatorname{argmin}_{\boldsymbol{\theta}} \ L_b(\boldsymbol{\theta}) + L_r(\boldsymbol{\theta}), \tag{2}$$

where $L_b(\boldsymbol{\theta}) = \frac{1}{N_b} \sum_{j=1}^{N_b} \left( \widehat{u}_{\boldsymbol{\theta}}(\mathbf{x}_b^j) - g(\mathbf{x}_b^j) \right)^2$ is the boundary term to fit the boundary condition, and $L_r(\boldsymbol{\theta}) = \frac{1}{N_c} \sum_{j=1}^{N_c} \left( \mathcal{F}[\widehat{u}_{\boldsymbol{\theta}}](\mathbf{x}_c^j) - f(\mathbf{x}_c^j) \right)^2$ is the residual term to fit the equation.

Despite many success stories, the PINN often struggles to solve PDEs with high-frequency and multi-scale components in the solutions. This is consistent with the "spectrum bias" observed in

---

[*]Equal contribution
[†]Corresponding author

NN training Rahaman et al. (2019). That is, NNs typically can learn the low-frequency information efficiently but grasping the high-frequency knowledge is much harder. To alleviate this problem, the recent work Wang et al. (2021b) proposes to construct a set of random Fourier features from zero-mean Gaussian distributions. The random features are then fed into the PINN layers for training (see (2)). While effective, the performance of this method is unstable, and is highly sensitive to the number and scales of the Gaussian variances, which are difficult to choose beforehand.

In this paper, we resort to an alternative arising ML solver framework, Gaussian processes (GP) (Chen et al., 2021; Long et al., 2022a). We propose GP-HM, a GP solver for High frequency and Multi-scale PDEs. By leveraging the Wiener-Khinchin theorem, we can directly model the solution in the frequency domain and estimate the target frequencies from the covariance function. We then develop an efficient learning algorithm to scale up to massive collocation points, which are critical to capture high frequencies. The major contributions of our work are as follows.

- **Model**. To flexibly capture the dominant frequencies, we use a mixture of student $t$ or Gaussian distributions to model the power spectrum of the solution. According to the Wiener-Khinchin theorem, we can derive the GP covariance function via inverse Fourier transform, which contains the component weights and frequency parameters. We show that by estimating the weights in the log domain, it is equivalent to assigning each weight a Jeffreys prior, which induces strong sparsity, automatically removes excessive frequency components, and drives the remaining toward the ground-truth. This way our GP can effectively extract the solution frequencies. Our covariance function derived from the Gaussian mixture power spectrum corresponds to the known spectral mixture kernel. We therefore are the first to realize its rationale and benefit for solving high-frequency and multi-scale PDEs.
- **Algorithm**. To enable efficient computation, we place all the collocation points and the boundary (and/or initial) points on a grid, and model the solution values at the grid with the GP finite projection. To obtain the derivative values in the equation, we compute the GP conditional mean via kernel differentiation. Next, we multiply our covariance function at each input dimension to obtain a product covariance. We then derive a Kronecker product form for the covariance and cross-covariance matrices. We use the properties of the Kronecker product and multilinear algebra to restrict the covariance matrix calculation to each input dimension. In this way, we can substantially reduce the cost and handle massive collocation points, without any low rank approximations.
- **Result**. We evaluated GP-HM with several benchmark PDEs that have high-frequency and multi-scale solutions. We compared with the standard PINN and several state-of-the-art variants. We compared with spectral methods (Boyd, 2001) that linearly combine a set of trigonometric bases to estimate the solution. We also compared with several other traditional numerical solvers. In all the cases, GP-HM consistently achieves relative $L_2$ errors at $\sim 10^{-3}$ or $\sim 10^{-4}$ or even smaller. By contrast, the competing ML based approaches often failed and gave much larger errors. The visualization of the element-wise prediction error shows that GP-HM also recovers the local solution values much better. We examined the learned frequency parameters, which match the ground-truth. Our ablation study as in Section C of Appendix also shows enough collocation points is critical to the success, implying the importance of our efficient learning method.

## 2 Gaussian Process

Gaussian processes (GPs) provide an expressive framework for function estimation. Suppose given a training dataset $\mathcal{D} = \{(\mathbf{x}_n, y_n) | 1 \leq n \leq N\}$, we aim to estimate a target function $f : \mathbb{R}^d \to \mathbb{R}$. We can assign a GP prior,

$$f(\cdot) \sim \mathcal{GP}(m(\cdot), \text{cov}(\cdot, \cdot)),$$

where $m(\cdot)$ is the mean function and $\text{cov}(\cdot, \cdot)$ is the covariance function. In practice, one often sets $m(\cdot) = 0$, and adopts a kernel function as the covariance function, namely $\text{cov}(f(\mathbf{x}), f(\mathbf{x}')) = k(\mathbf{x}, \mathbf{x}')$. A nice property of the GP prior is that if $f$ is sampled from a GP, then any derivative (if existent) of $f$ is also a GP, and the covariance between the derivative and the function $f$ is the derivative of the kernel function w.r.t the same input variable(s). For example,

$$\text{cov}(\partial_{x_1 x_2} f(\mathbf{x}), f(\mathbf{x}')) = \partial_{x_1 x_2} k(\mathbf{x}, \mathbf{x}'), \tag{3}$$

where $\mathbf{x} = (x_1, \ldots, x_d)^\top$ and $\mathbf{x}' = (x_1', \ldots, x_d')^\top$. Under the GP prior, the function values at any finite input collection, $\mathbf{f} = [f(\mathbf{x}_1), \ldots, f(\mathbf{x}_N)]$, follow a multi-variate Gaussian distribution, $p(\mathbf{f}) = \mathcal{N}(\mathbf{f}|\mathbf{0}, \mathbf{K})$ where $[\mathbf{K}]_{ij} = \text{cov}(f(\mathbf{x}_i), f(\mathbf{x}_j)) = k(\mathbf{x}_i, \mathbf{x}_j)$. This is called a GP projection. Suppose given $\mathbf{f}$, we want to compute the distribution of the function value at any input $\mathbf{x}$, namely $p(f(\mathbf{x})|\mathbf{f})$. Since $\mathbf{f}$ and $f(\mathbf{x})$ also follow a multi-variate Gaussian distribution, we obtain a conditional Gaussian, $p(f(\mathbf{x})|\mathbf{f}) = \mathcal{N}(f(\mathbf{x})|\mu(\mathbf{x}), \sigma^2(\mathbf{x}))$, where the conditional mean

$$\mu(\mathbf{x}) = \text{cov}(f(\mathbf{x}), \mathbf{f})\mathbf{K}^{-1}\mathbf{f}, \tag{4}$$

and $\sigma^2(\mathbf{x}) = \text{cov}(f(\mathbf{x}), f(\mathbf{x})) - \text{cov}(f(\mathbf{x}), \mathbf{f})\mathbf{K}^{-1}\text{cov}(\mathbf{f}, f(\mathbf{x}))$, $\text{cov}(f(\mathbf{x}), \mathbf{f}) = k(\mathbf{x}, \mathbf{X}) = [k(\mathbf{x}, \mathbf{x}_1), \ldots, k(\mathbf{x}, \mathbf{x}_N)]$ and $\mathbf{X} = [\mathbf{x}_1, \ldots, \mathbf{x}_N]^\top$.

## 3 Gaussian Process PDE Solvers

**Covariance Design.** When the PDE solution $u$ includes high frequencies or multi-scale information, one naturally wants to estimate these target frequencies outright in the frequency domain. To this end, we consider the solution's power spectrum, $S(s) = |\hat{u}(s)|^2$ where $\hat{u}(s)$ is the Fourier transform of $u$, and $s$ denotes the frequency. The power spectrum characterizes the strength of every possible frequency within the solution. To flexibly capture the dominant high and/or multi-scale frequencies, we use a mixture of student $t$ distributions to model the power spectrum,

$$S(s) = \sum\nolimits_{q=1}^{Q} w_q \text{St}(s; \mu_q, \rho_q^2, \nu), \tag{5}$$

where $w_q > 0$ is the weight of component $q$, St stands for student $t$ distribution, $\mu_q$ is the mean, $\rho_q^2$ is the inverse variance, and $\nu$ is the degree of freedom. Note that $w_q$ does not need to be normalized (their summation is not necessary to be one). Each student $t$ distribution characterizes one principle frequency $\mu_q$, and also robustly models the (potentially many) minor frequencies with a fat tailed density (Bishop, 2007). An alternative choice is a mixture of Gaussian, $S(s) = \sum_{q=1}^{Q} w_q \mathcal{N}(s; \mu_q, \rho_q^2)$. But the Gaussian distribution has thin tails, hence is sensitive to long-tail outliers and can be less robust (in capturing minor frequencies).

Next, we convert the spectrum model into a covariance function to enable our GP solver to flexibly estimate the target frequencies. According to the Wiener-Khinchin theorem (Wiener, 1930; Khintchine, 1934), for a wide-sense stationary random process, under mild conditions, its power spectrum[1] and the auto-correlation form a Fourier pair. We model the solution $u$ as drawn from a stationary GP, and the auto-correlation is the covariance function, denoted by $k(x, x') = k(x - x')$. We then have

$$S(s) = \int k(z)e^{-i2\pi sz}\mathrm{d}z, \quad k(z) = \int S(s)e^{i2\pi zs}\mathrm{d}s, \tag{6}$$

where $z = x - x'$, and $i$ indicates complex numbers. Therefore, we can obtain the covariance function by applying the inverse Fourier transform over $S(s)$. However, the straightforward mixture in (5) will lead to a complex-valued covariance function. To obtain a real-valued covariance, inside each component we add another student $t$ distribution with mean $-u_q$ so as to cancel out the imaginary part after integration. In addition, to make the derivation convenient, we scale the inverse variance and degree of freedom by a constant. We use the following power spectrum model,

$$S(s) = \sum\nolimits_{q=1}^{Q} w_q \left( \text{St}(s; \mu_q, 4\pi^2\rho_q^2, 2\nu) + \text{St}(s; -\mu_q, 4\pi^2\rho_q^2, 2\nu) \right). \tag{7}$$

Applying inverse Fourier transform in (6), we can derive the following covariance function,

$$k_{\text{StM}}(x, x') = \sum\nolimits_{q=1}^{Q} w_q \gamma_{\nu, \rho_q}(x, x') \cos(2\pi\mu_q(x - x')), \tag{8}$$

where $\gamma_{\nu, \rho_q}(x, x') = \frac{2^{1-\nu}}{\Gamma(\nu)} \left( \sqrt{2\nu} \frac{|x-x'|}{\rho_q} \right)^\nu K_\nu(\sqrt{2\nu} \frac{|x-x'|}{\rho_q})$ is the Matérn kernel with degree of freedom $\nu$ and length scale $\rho_q$, and $K_\nu$ is the modified Bessel function of the second kind. The details of the derivation is left in Appendix. We now can see that the frequency information $\mu_q$ and

---

[1]To be well-posed, the power spectrum for a random process is defined in a slightly different way (taking the limit of a windowed signal), but it reflects the same insight; see (Lathi, 1998; Grimmett and Stirzaker, 2020) for details.

component weights $w_q$ are embedded into the covariance function. By learning a GP model, we expect to capture the true frequencies of the solution. One can also construct a symmetric Gaussian mixture in the same way, and via inverse Fourier transform obtain

$$k_{\text{GM}}(x, x') = \sum\nolimits_{q=1}^{Q} w_q \exp\left(-\rho_q^2(x - x')^2\right) \cdot \cos(2\pi(x - x')\mu_q). \tag{9}$$

This is known as the spectral mixture kernel (Wilson and Adams, 2013), which was originally proposed to construct an expressive stationary kernel according to its Fourier decomposition, because in principle the Gaussian mixture can well approximate any distribution, as long as using enough many components. Wilson and Adams (2013) showed that the spectral mixture kernel can well recover many popular kernels, such as rational quadratic and periodic kernel. In this paper, we take a different motivation and viewpoint. We argue that a similar design can be very effective in extracting dominant frequencies in PDE solving.

**How to Determine the Component Number?** Since the number of dominant frequencies is unknown apriori, the solution accuracy can be sensitive to the choice of the component number $Q$. A too small $Q$ can miss important (high) frequencies while a too big $Q$ can bring in excessive noisy frequencies. To address this problem, we set a large $Q$ (e.g., 50), initialize the frequency parameters $\mu_q$ across a wide range, and then optimize the component weights in the log domain. This turns out to be equivalent to assigning each $w_q$ a Jefferys prior. Specifically, define $\overline{w}_q = \log(w_q)$. Since we do not place an additional prior over $\overline{w}_q$, we can view $p(\overline{w}_q) \propto 1$. Then we have

$$p(w_q) = p(\overline{w}_q)\left|\frac{\mathrm{d}\overline{w}_q}{\mathrm{d}w_q}\right| \propto \frac{1}{w_q}. \tag{10}$$

The Jeffreys prior has a very high density near zero, and hence induces strong sparsity during the learning of $w_q$ (Figueiredo, 2001). Accordingly, the excessive frequency components can be automatically pruned, and the learning drives the remaining $\mu_q$'s toward the target frequencies. This have been verified by our experiments; see Fig. 4 in Section 6.

**GP Solver Model to Enable Fast Computation.** To fulfill efficient and scalable calculation, we multiply our covariance function at each input dimension to construct a product kernel,

$$\text{cov}(f(\mathbf{x}), f(\mathbf{x}')) = \kappa(\mathbf{x}, \mathbf{x}'|\Theta) = \prod\nolimits_{j=1}^{d} k_{\text{StM}}(x_j, x_j'|\boldsymbol{\theta}_q), \tag{11}$$

where $\boldsymbol{\theta}_q = \{w_q, \mu_q, \rho_q\}$ and $\Theta = \{\boldsymbol{\theta}_q\}_{q=1}^{Q}$ are the kernel parameters. Note that the product kernel is equivalent to performing a (high-dimensional) feature mapping for each input dimension and then computing the tensor-product across the features. It is a highly expressive structure and commonly used in finite element design (ARNOLD et al., 2012). Next, we create a grid on the domain $\Omega$. We can randomly sample or specially design the locations at each input dimension, and then construct the grid through a Cartesian product. Denote the locations at each input dimension $j$ by $\mathbf{h}_j = [h_{j1}, \ldots, h_{jM_j}]$, we have an $M_1 \times \ldots \times M_d$ grid,

$$\mathcal{G} = \mathbf{h}_1 \times \ldots \times \mathbf{h}_d = \{\mathbf{x} = (x_1, \ldots, x_d)|x_j \in \mathbf{h}_j, 1 \leq j \leq d\}. \tag{12}$$

We will use the grid points on the boundary $\partial\Omega$ to fit the boundary conditions and all the grid points as the collocation points to fit the equation.

Denote the solution values at $\mathcal{G}$ by $\mathcal{U} = \{u(\mathbf{x})|\mathbf{x} \in \mathcal{G}\}$, which is an $M_1 \times \ldots \times M_d$ array. According to the GP prior over $u(\cdot)$, we have a multi-variate Gaussian prior distribution, $p(\mathcal{U}) = \mathcal{N}(\text{vec}(\mathcal{U})|\mathbf{0}, \mathbf{C})$, where $\text{vec}(\cdot)$ is to flatten $\mathcal{U}$ into a vector, $\mathbf{C}$ is the covariance matrix computed from $\mathcal{G}$ with kernel $\kappa(\cdot, \cdot)$. Denote the grid points on the boundary by $\mathcal{B} = \mathcal{G} \cap \partial\Omega$. To fit the boundary condition, we use a Gaussian likelihood, $p(\mathbf{g}|\mathbf{u}_\mathcal{B}) = \mathcal{N}(\mathbf{g}|\mathbf{u}_b, \tau_1^{-1}\mathbf{I})$, where $\mathbf{g} = \text{vec}\left(\{g(\mathbf{x})|\mathbf{x} \in \mathcal{B}\}\right)$, $\mathbf{u}_b$ are the values of $\mathcal{U}$ on $\mathcal{B}$ (flatten into a vector), and $\tau_1 > 0$ is the inverse variance. Next, we want to fit the equation at $\mathcal{G}$. To this end, we need to first obtain the prediction of all the relevant derivatives of $u$ in the PDE, e.g., $\partial_{x_1} u$ and $\partial_{x_1 x_2} u$, at the grid $\mathcal{G}$. Since $u$'s derivatives also follow the GP prior, we use the kernel derivative to obtain their cross covariance function (see (3)), with which to compute the GP conditional mean (conditioned on $\mathcal{U}$) as the prediction. Take $\partial_{x_1} u$ and $\partial_{x_1 x_2} u$ as examples. We have

$$\partial_{x_1} u(\mathbf{x}) = \partial_{x_1}\mathbf{k}(\mathbf{x}, \mathcal{G})\mathbf{C}^{-1}\text{vec}(\mathcal{U}), \quad \partial_{x_1 x_2} u(\mathbf{x}) = \partial_{x_1 x_2}\mathbf{k}(\mathbf{x}, \mathcal{G})\mathbf{C}^{-1}\text{vec}(\mathcal{U}), \tag{13}$$

where $\mathbf{k}(\mathbf{x}, \mathcal{G}) = [k(\mathbf{x}, \mathbf{x}_1'), \ldots, k(\mathbf{x}, \mathbf{x}_M')]$ where $M = \prod_j M_j$ and all $\mathbf{x}_j'$ constitute $\mathcal{G}$. We can accordingly predict the values of the all the relevant $u$ derivatives at $\mathcal{G}$, and combine them to obtain

the PDE (see (1)) evaluation at $\mathcal{G}$, which we denote by $\mathcal{H}$. To fit the GP model to the equation, we use another Gaussian likelihood, $p(\mathbf{0}|\mathcal{U}) = \mathcal{N}(\mathbf{0}|\text{vec}(\mathcal{H}), \tau_2^{-1}\mathbf{I})$, where $\mathbf{0}$ is an virtual observation, and $\tau_2 > 0$. Note that we use the same framework as in (Chen et al., 2021; Long et al., 2022b). However, there are two critical differences. First, rather than randomly sample the collocation points, we place all the collocation points on a grid. Second, rather than assign a multivariate Gaussian distribution over the function values and all of its derivatives, we only model the distribution of the function values (at the grid). We then use the GP conditional mean to predict the derivative values. As we will discuss in Section 4, these modeling strategies, coupled with the product covariance (11), enable highly efficient and scalable computation, yet do not need any low rank approximations.

## 4 Algorithm

We maximize the log joint probability[2] to estimate $\mathcal{U}$, the kernel parameters $\Theta$, and the likelihood inverse variances $\tau_1$ and $\tau_2$. To flexibly adjust the influence of the boundary likelihood so as to balance the competition between the boundary and equation likelihoods (Wang et al., 2020a;c), we introduce a free hyper-parameter $\lambda_b > 0$, and maximize the weighted log joint probability,

$$
\begin{aligned}
\mathcal{L}(\mathcal{U}, \Theta, \tau_1, \tau_2) = & \log \mathcal{N}(\text{vec}(\mathcal{U})|\mathbf{0}, \mathbf{C}) + \lambda_b \cdot \log \mathcal{N}(\mathbf{g}|\mathbf{u}_b, \tau_1^{-1}\mathbf{I}) + \log \mathcal{N}(\mathbf{0}|\text{vec}(\mathcal{H}), \tau_2^{-1}\mathbf{I}) \\
= & -\frac{1}{2}\log|\mathbf{C}| - \frac{1}{2}\text{vec}(\mathcal{U})^\top \mathbf{C}^{-1}\text{vec}(\mathcal{U}) + \lambda_b \left[ \frac{N_b}{2}\log\tau_1 - \frac{\tau_1}{2}\|\mathbf{u}_b - \mathbf{g}\|^2 \right] \\
& + \frac{M}{2}\log\tau_2 - \frac{\tau_2}{2}\|\text{vec}(\mathcal{H})\|^2 + \text{const.}
\end{aligned}
\tag{14}
$$

Naive computation of $\mathcal{L}$ is extremely expensive when the grid is dense, namely, $M$ is large. That is because the covariance matrix $\mathbf{C}$ is between all the grid points, of size $M \times M$ ($M = \prod_j M_j$). Also, to obtain $\mathcal{H}$, we need to compute the cross-covariance between every needed derivative in the PDE and $u$ across all the grid points. Consequently, the naive computation of the log determinant and inverse of $\mathbf{C}$ (see (14) ) and the required cross-covariance take the time and space complexity $\mathcal{O}(M^3)$ and $\mathcal{O}(M^2)$, respectively, which can be infeasible even when each $M_j$ is relatively small. For example, when $d = 3$, and $M_1 = M_2 = M_3 = 100$, we have $M = 10^6$ and the computation of $\mathbf{C}$ will be too costly to be practical (on most computing platforms).

Thanks to that (1) our prior distribution is only on all the function values at the grid, and (2) our covariance function is a product over each input dimension (see (11)). We can derive a Kronecker product structure in $\mathbf{C}$, namely, $\mathbf{C} = \mathbf{C}_1 \otimes \ldots \otimes \mathbf{C}_d$, where $C_j = k_{\text{StM}}(\mathbf{h}_j, \mathbf{h}_j)$ is the kernel matrix on $\mathbf{h}_j$ — the locations at input dimension $j$, of size $M_j \times M_j$. Note that we can also use $k_{\text{GM}}$ in (9). Using the Kronecker product properties (Minka, 2000), we obtain

$$
\log|\mathbf{C}| = \sum_{j=1}^{d} \frac{M}{M_j}\log|\mathbf{C}_j|,
$$

$$
\mathbf{C}^{-1}\text{vec}(\mathcal{U}) = \left(\mathbf{C}_1^{-1} \otimes \ldots \otimes \mathbf{C}_d^{-1}\right)\text{vec}(\mathcal{U}) = \text{vec}\left(\mathcal{U} \times_1 \mathbf{C}_1^{-1} \times_2 \ldots \times_d \mathbf{C}_d^{-1}\right),
\tag{15}
$$

where $\times_j$ is the tensor-matrix product at mode $j$. Accordingly, we can first compute the local log determinant and inverse at each input dimension (i.e., for each $\mathbf{C}_j$), which reduces the time and space complexity to $\mathcal{O}(\sum_{j=1}^{d} M_j^3)$ and $\mathcal{O}(\sum_{j=1}^{d} M_j^2)$, respectively. Then we perform the multilinear operation in the last line of (15), i.e., sequentially multiplying the array $\mathcal{U}$ with each $C_j^{-1}$, which takes the time complexity $\mathcal{O}\left((\sum_{j=1}^{d} M_j)M\right)$. The computational cost is substantially reduced.

Furthermore, since our product covariance function is factorized over each input dimension, the cross covariance between any derivative of $u$ and $u$ itself still maintains a product form — because only the kernel(s) at the corresponding input dimension(s) need to be differentiated. For example,

$$
\begin{aligned}
\text{cov}(\partial_{x_1 x_2} u(\mathbf{x}), u(\mathbf{x}')) &= \partial_{x_1 x_2}\kappa(\mathbf{x}, \mathbf{x}') = \partial_{x_1 x_2}\prod_j \kappa(x_j, x_j') \\
&= \partial_{x_1}\kappa(x_1, x_1') \cdot \partial_{x_2}\kappa(x_2, x_2') \cdot \prod_{j \neq 1, 2}\kappa(x_j, x_j').
\end{aligned}
\tag{16}
$$

---

[2]We found that performing posterior inference over $\mathcal{U}$ and other parameters, e.g., via variational inference, will degrade the solution accuracy, which partly be because the inference and optimization is much more complicated and challenging.

Accordingly, we can also obtain Kronecker product structures in predicting each derivative of $u$. Take $\partial_{x_1 x_2} u$ as an example. According to (13), we can derive that

$$\partial_{x_1 x_2} u(\mathbf{x}) = \left(\partial_{x_1} \mathbf{k}(x_1, \mathbf{h}_1) \otimes \partial_{x_2} \mathbf{k}(x_2, \mathbf{h}_2) \otimes \ldots \otimes \mathbf{k}(x_d, \mathbf{h}_d)\right) \left(\mathbf{C}_1^{-1} \otimes \ldots \otimes \mathbf{C}_d^{-1}\right) \text{vec}(\mathcal{U})$$

$$= \left(\partial_{x_1} \mathbf{k}(x_1, \mathbf{h}_1)\mathbf{C}_1^{-1} \otimes \partial_{x_2} \mathbf{k}(x_2, \mathbf{h}_2)\mathbf{C}_2^{-1} \otimes \ldots \otimes \mathbf{k}(x_d, \mathbf{h}_d)\mathbf{C}_d^{-1}\right) \text{vec}(\mathcal{U})$$

$$= \text{vec}\left(\mathcal{U} \times_1 \partial_{x_1} \mathbf{k}(x_1, \mathbf{h}_1)\mathbf{C}_1^{-1} \times_2 \partial_{x_2} \mathbf{k}(x_2, \mathbf{h}_2)\mathbf{C}_2^{-1} \times_3 \mathbf{k}(x_3, \mathbf{h}_3)\mathbf{C}_3^{-1} \times_4 \ldots \times_d \mathbf{k}(x_d, \mathbf{h}_d)\mathbf{C}_d^{-1}\right).$$

Denote the values of $\partial_{x_1 x_2} u$ at the grid $\mathcal{M}$ by $\partial_{x_1 x_2} \mathcal{U} \triangleq \{\partial_{x_1 x_2} u(\mathbf{x}) | \mathbf{x} \in \mathcal{M}\}$. Then it is straightforward to obtain $\partial_{x_1 x_2} \mathcal{U} = \mathcal{U} \times_1 \nabla_1 \mathbf{C}_1 \mathbf{C}_1^{-1} \times_2 \nabla_1 \mathbf{C}_2 \mathbf{C}_2^{-1}$, where $\nabla_1$ means taking the derivative w.r.t the first input variable, and we have $\nabla_1 \mathbf{C}_1 = [\partial_{h_{11}} \mathbf{k}(h_{11}, \mathbf{h}_1); \ldots; \partial_{h_{1M_1}} \mathbf{k}(h_{1M_1}, \mathbf{h}_1)]$ and $\nabla_1 \mathbf{C}_2 = [\partial_{h_{21}} \mathbf{k}(h_{21}, \mathbf{h}_2); \ldots; \partial_{h_{2M_2}} \mathbf{k}(h_{2M_2}, \mathbf{h}_2)]$. Hence, we just need to perform two tensor-matrix products, which takes $\mathcal{O}((M_1 + M_2)M)$ operations, and is efficient and convenient. Similarly, we can compute the prediction of all the associated $u$ derivatives in the PDE operator, with which we can obtain $\mathcal{H}$ — the PDE evaluation at the grid in (14). We can then use automatic differentiation to calculate the gradient to maximize (14).

**Algorithm Complexity.** The time complexity of our algorithm is $\mathcal{O}(\sum_j M_j^3 + (\sum_j M_j)M)$. The space complexity is $\mathcal{O}(\sum_j M_j^2 + M)$, including the storage of the covariance matrix at each input dimension, and the solution estimate at grid $\mathcal{G}$, namely $\mathcal{U}$.

## 5 Related Work

Although the PINN has many success stories, e.g., (Raissi et al., 2020; Chen et al., 2020; Jin et al., 2021; Sirignano and Spiliopoulos, 2018; Zhu et al., 2019; Geneva and Zabaras, 2020; Sahli Costabal et al., 2020), the training is known to be challenging, which is partly due to that applying differential operators over the NN can complicate the loss landscape (Krishnapriyan et al., 2021). Recent works have analyzed common failure modes of PINNs which include modeling problems exhibiting high-frequency, multi-scale, chaotic, or turbulent behaviors (Wang et al., 2020c;b;a; 2022), or when the governing PDEs are stiff (Krishnapriyan et al., 2021; Mojgani et al., 2022). One class of approaches to mitigate the training challenge is to set different weights for the boundary and residual loss terms. For example, Wight and Zhao (2020) suggested to set a large weight for the boundary loss to prevent the dominance of the residual loss. Wang et al. (2020a) proposed a dynamic weighting scheme based on the gradient statistics of the loss terms. Wang et al. (2020c) developed an adaptive weighting approach based on the eigen-values of NTK. Liu and Wang (2021) employed a mini-max optimization and updated the loss weights via stochastic ascent. McClenny and Braga-Neto (2020) used a multiplicative soft attention mask to dynamically re-weight the loss term on each data point and collocation point. Another strategy is to modify the NN architecture so as to exactly satisfy the boundary conditions, e.g., (Lu et al., 2021; Lyu et al., 2020; Lagaris et al., 1998). However, these methods are restricted to particular types of boundary conditions, and are less flexible than the original PINN framework. Tancik et al. (2020); Wang et al. (2021b) used Gaussian distributions to construct random Fourier features to improve the learning of the high-frequency and multi-scale information. The number of Gaussian variances and their scales are critical to the success of these methods. But these hyperparameters are quite difficult to choose.

Earlier works (Graepel, 2003) have used GP for solving linear PDEs with noisy measurement of source terms. In (Wang et al., 2021a), the rationale and guarantees of using GP as a prior for PDE solutions are discussed. The work also justifies the usage of the product kernel in terms of sample path properties. The recent work (Chen et al., 2021) develops a general approach for solving both linear and nonlinear PDEs. Long et al. (2022b) proposed a GP framework to integrate various differential equations. The recent work (Chen et al., 2023) uses sparse inverse Cholesky factorization to approximate the kernel matrix so as to handle a large number of collocation points. These methods use SE and Matérn kernels and are challenging to capture high-frequency and multi-scale solutions. The recent work (Pförtner et al., 2022) proposes a physics-informed GP solver for linear PDEs that generalizes weighted residuals. In (Härkönen et al., 2022), a GP kernel is constructed via the Ehrenpreis-Palamodov fundamental principle and nonlinear Fourier transform to solve linear PDEs with constant coefficients. This work also derives the spectral mixture kernel as an instance of its own kernel design. The computational advantage of using Kronecker product structures have been realized in (Saatcci, 2012), and applied in other tasks, such as nonparametric tensor decomposition (Xu et al., 2012), sparse approximation with massive inducing points (Wilson and Nickisch, 2015; Izmailov

et al., 2018), and high-dimensional output regression (Zhe et al., 2019). In Wilson et al. (2015) it further points out that if one uses a regular (evenly-spaced), each kernel matrix will has a Toeplitz structure, which can lead to $O(n \log n)$ computation. However, in machine learning applications, data is typically not observed at a grid and the Kronecker product has a limited usage. By contrast, for PDE solving, it is natural to estimate the solution values on a grid, which opens the possibility of using Kronecker products combined with GP for efficient computation. More general discussions about Bayesian learning and PDE problems are given in (Owhadi, 2015; Cockayne et al., 2017). Tensor methods used in numerical computation are discussed in (Gavrilyuk and Khoromskij, 2019).

# 6 Experiment

To evaluate GP-HM, we considered three commonly-used benchmark PDE families in the literature of machine learning solvers (Raissi et al., 2019; Wang et al., 2021b; Krishnapriyan et al., 2021): *Poisson*, *Allen-Cahn* and *Advection*. Following the prior works, we fabricated a series of solutions to thoroughly examine the performance. The details are given in Section B of Appendix.

We compared with the following state-of-the-art ML solvers: (1) standard PINN, (2) Weighted PINN (W-PINN) that up-weight the boundary loss to reduce the dominance of the residual loss, and to more effectively propagate the boundary information, (3) Rowdy (Jagtap et al., 2022), PINN with an adaptive activation function, which combines a standard activation with several $\sin$ or $\cos$ activations. (4) RFF-PINN, feeding Random Fourier Features to the PINN (Wang et al., 2021b). To ensure RFF-PINN to achieve the best performance, we followed (Wang et al., 2020c) to dynamically re-weight the loss terms based on NTK eigenvalues (Wang et al., 2020). (5) Spectral Method (Boyd, 2001), which approximates the solution with a linear combination of trigonometric bases, and estimates the basis coefficients via least mean squares. In addition, we also tested (6) GP-SE and (7) GP-Matérn, GP solvers with the square exponential (SE) and the Matérn kernel. The details about the hyperparameter setting and tuning is provided in Section B of Appendix. We denote our method using the covariance function based on (8) and (9) by GP-HM-StM and GP-HM-GM, respectively. We compared with several traditional numerical solvers: (8) Chebfun[3] that solves PDEs based on Chebyshev interpolants, (9) Finite Difference (FD), which solves the PDEs via discretization based on finite difference. We used PyPDE library[4] to solve 1D/2D Poisson equations, and 1D advection (using methods of lines). Note that PyPDE does not support solving nonlinear stationary PDEs, namely 1D/2D Allen-Cahn Equation in (28), and so we implemented the finite difference with Scipy and Krylov method for root finding. Note also that the *Chebfun library does not support 2D Poisson and nonlinear stationary PDEs, namely, 1D/2D Allen-Cahn equation, and so it has very limited usage*. We employed the default settings in Chebfun library. When using PyPDE, we set spacial discretization to 400 and 400 time steps (if needed). For 1D Allen-cahn, the spatial discretization is set to 400. For 2D Allen-cahn, we used a $45 \times 45$ grid; otherwise, the root finding either ran forever or failed due to numerical instability. We have also tested the Spectral Galerking method implemented by the Shenfun library[5]. However, we found it failed in every test case (the relative $L_2$ error is at several thousands). Hence, we did not report the results.

**Solution Accuracy.** We report the relative $L_2$ error (normalized root-mean-square error) of each method in Table 1 and 2. The best result and the smaller error between GP-HM-StM and GP-HM-GM are made bold. We can see that, *among all the ML solvers*, our method achieves the smallest solution error in all the cases except that for the 1D Poisson equation with solution $u_2$, RFF-PINN is better. However, in all the cases, the solution error of GP-HM achieves at least 1e-3 level. In quite a few cases, our method even reaches an error around 1e-6 and 1e-7. It shows that GP-HM can successfully solve all these equations. By contrast, GP solvers using the plain SE and Matérn kernel result in several orders of the magnitude bigger errors. The standard PINN and W-PINN basically failed to solve every equation. While Rowdy improved upon PINN and W-PINN in most cases, the error is still quite large. The inferior performance of the spectral method implies that only using trigonometric bases is *not* sufficient. With the usage of the random Fourier features, RFF-PINN can greatly boost the performance of PINN and W-PINN in many cases. However, in most cases, it is still much inferior to GP-HM. The performance of RFF-PINN is very sensitive to the number and scales of the Gaussian variance, and these hyper-parameters are not easy to choose. We have tried 20

---

[3]`https://www.chebfun.org/`
[4]`https://py-pde.readthedocs.io/en/latest/`
[5]`https://shenfun.readthedocs.io/en/latest/`

| Method | 1D | | | | | 2D | |
|---|---|---|---|---|---|---|---|
| | $u_1$ | $u_2$ | $u_3$ | $u_4$ | $u_5$ | $u_6$ | $u_7$ |
| PINN | 1.36e0 | 1.40e0 | 1.00e0 | 1.42e1 | 6.03e-1 | 1.63e0 | 9.99e-1 |
| W-PINN | 1.31e0 | 2.65e-1 | 1.86e0 | 2.60e1 | 6.94e-1 | 1.63e0 | 6.75e-1 |
| RFF-PINN | 4.97e-4 | 2.00e-5 | 7.29e-2 | 2.80e-1 | 5.74e-1 | 1.69e0 | 7.99 e-1 |
| Rowdy | 1.70e0 | 1.00e0 | 1.00e0 | 1.01e0 | 1.03e0 | 2.24e1 | 7.36e-1 |
| Spectral method | 2.36e-2 | 3.47e0 | 1.02e0 | 1.02e0 | 9.98e-1 | 1.58e-2 | 1.04e0 |
| Chebfun | **3.05e-11** | **1.17e-11** | **5.81e-11** | **1.14e-10** | **8.95e-10** | N/A | N/A |
| Finite Difference | 5.58e-1 | 4.78e-2 | 2.34e-1 | 1.47e0 | 1.40e0 | 2.33e-1 | 1.75e-2 |
| GP-SE | 2.70e-2 | 9.99e-1 | 9.99e-1 | 3.19e-1 | 9.75e-1 | 9.99e-1 | 9.53e-1 |
| GP-Matérn | 3.32e-2 | 9.8e-1 | 5.15e-1 | 1.83e-2 | 6.27e-1 | 6.28e-1 | 3.54e-2 |
| GP-HM-GM | **3.99e-7** | 2.73e-3 | 3.92e-6 | 1.55e-6 | 1.82e-3 | **6.46e-5** | 1.06e-3 |
| GP-HM-StM | 6.53e-7 | **2.71e-3** | **3.17e-6** | **8.97e-7** | **4.22e-4** | 6.87e-5 | **1.02e-3** |

Table 1: Relative $L_2$ error in solving 1D and 2D Poisson equations, where $u_j$'s are different high-frequency and multi-scale solutions: $u_1 = \sin(100x)$, $u_2 = \sin(x) + 0.1\sin(20x) + 0.05\cos(100x)$, $u_3 = \sin(6x)\cos(100x)$, $u_4 = x\sin(200x)$, $u_5 = \sin(500x) - 2(x - 0.5)^2$, $u_6 = \sin(100x)\sin(100y)$ and $u_7 = \sin(6x)\sin(20x) + \sin(6y)\sin(20y)$.

| Method | 1D Allen-cahn | | 2D Allen-cahn | 1D Advection |
|---|---|---|---|---|
| | $u_1$ | $u_2$ | | |
| PINN | 1.41e0 | 1.14e1 | 1.96e1 | 1.00e0 |
| W-PINN | 1.34e0 | 1.45e1 | 2.03e1 | 1.01e0 |
| RFF-PINN | 1.24e-3 | 2.46e-1 | 7.17e-1 | 9.96e-1 |
| Rowdy | 1.30e0 | 1.31e0 | 1.18e0 | 1.03e0 |
| Spectral method | 2.34e-2 | 2.45e1 | 2.45e1 | 2.67e0 |
| Chebfun | **1.39e-08** | **2.94e-10** | N/A | 1.39e0 |
| Finite Difference | 2.32e-01 | 2.36e-1 | 3.23e0 | 1.29e-1 |
| GP-SE | 2.74e-2 | 1.06e-2 | 3.48e-1 | 9.99e-1 |
| GP-Matérn | 3.32e-2 | 5.16e-2 | 2.96e-1 | 9.99e-1 |
| GP-HM-StM | 7.71e-6 | 4.76e-6 | **2.99e-3** | **9.08e-4** |
| GP-HM-GM | **4.91e-6** | **4.24e-6** | 5.78e-3 | 3.59e-3 |

Table 2: Relative $L_2$ error in solving 1D, 2D Allen-cahn equations and 1D advection equation, where $u_1$ and $u_2$ are two test solutions for 1D Allen-cahn: $u_1 = \sin(100x)$, $u_2 = \sin(6x)\cos(100x)$. The test solution for 2D Allen-cahn is $(\sin(x) + 0.1\sin(20x) + \cos(100x)) \cdot (\sin(y) + 0.1\sin(20y) + \cos(100y))$, and for 1D advection equation is $\sin(x - 200t)$.

settings and report the best performance (see Section B in Appendix). Compared with traditional solvers, we can see Chebfun performs very well, and achieves the highest solution accuracy except for the 1D advection problem. However, Chebfun is limited to 1D problems and temporal PDEs. It cannot handle 2D stationary PDEs, no matter linear or nonlinear. Finite Difference can provide reasonable accuracy, but the performance is consistently much worse than GP-HM. This might be due to the challenge in solving the root finding problem, caused by the high-frequency/multi-scale information implied in the source term. Overall, we can see that our method is general enough to solve different types of PDEs (1D/2D, linear/nonlinear, stationary and non-stationary); to achieve a satisfactory accuracy, we do NOT need to change the computation framework to re-develop the solver. By contrast, it is known that the success of numerical solvers tightly binds to the specific problem, domain knowledge, skillful implementation, and numerous numerical tricks. Any change of these aspects can cause failures of the solvers and demand for a re-design and re-implementation. It therefore brings significant challenges in usage.

**Point-wise Error.** We then show the point-wise solution error in Fig. 1, 2, 3, and in Appendix Fig. 5, 6, 7. We can see that GP-SE is difficult to capture high frequencies. While GP-Matérn is better, it is unable to grasp all the scale information. RFF-PINN successfully captured multi-scale frequencies in Fig. 1, but it failed in more challenging cases as in Fig. 2 and 3. In 2D Poisson and 1D Advection, the point-wise error of both GP-HM-StM and GP-HM-GM is quite uniform across the domain and is close to zero (dark blue); see Fig. 3, and in Appendix Fig. 6, 7. By contrast, the other methods exhibit large errors in a few local regions. These results show that GP-HM not only gives a superior global accuracy, but locally recovers individual solution values.

**Frequency Learning.** Third, we investigated the learned component weights $w_q$ and frequencies $\mu_q$ of GP-HM. In Fig. 4, we show the results for two Poisson equations. As we can see, although the number of components $Q$ is set to be much larger than the number of true frequencies, the estimation

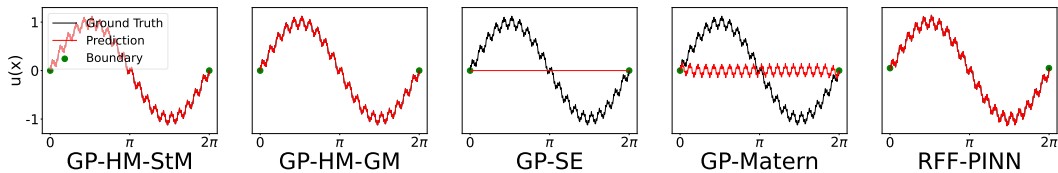

Figure 1: Prediction for the 1D Poisson equation with solution $\sin(x) + 0.1\sin(20x) + 0.05\cos(100x)$.

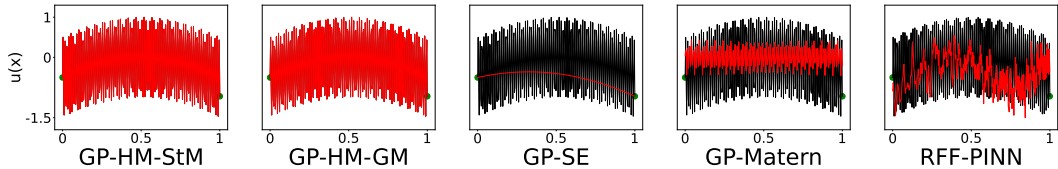

Figure 2: Prediction for the 1D Poisson equation with solution $\sin(500x) - 2(x - 0.5)^2$.

of most weights $w_q$ is very small (less than $10^{-10}$). That means, excessive frequency components have been automatically pruned. The remaining components with significant weights completely match the number of true frequencies in the solution. The frequency estimation $\mu_q$ is very close to the ground-truth. This demonstrates that the implicit Jefferys prior (by optimizing $w_q$ in the log space) can indeed implement sparsity, select the right frequency number, and recover the ground-truth frequency values. Finally, we show additional results in Section C of Appendix.

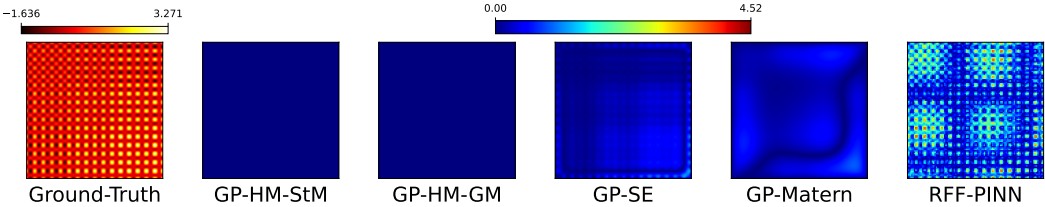

Figure 3: Point-wise solution error for 2D Allen-cahn equation, and the solution is $(\sin(x) + 0.1\sin(20x) + \cos(100x))(\sin(y) + 0.1\sin(20y) + \cos(100y))$.

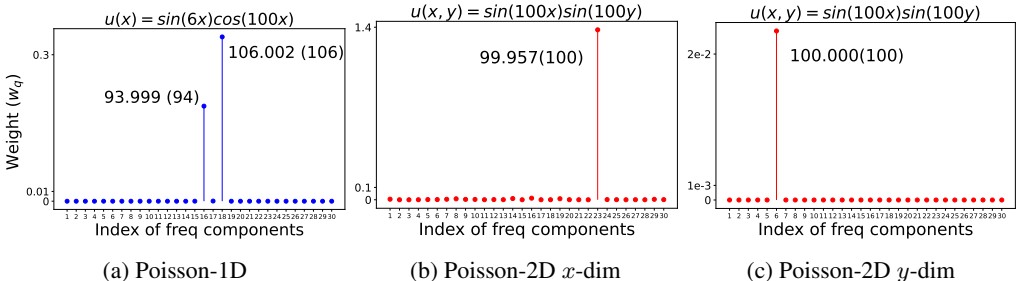

(a) Poisson-1D  (b) Poisson-2D $x$-dim  (c) Poisson-2D $y$-dim

Figure 4: The learned component weights and frequency values. For each number pair a(b) in the figure, "a" is the learned frequency by GP-HM, and "b" is the ground-truth. The expressions on the top are the solutions.

## 7 Conclusion

We have presented GP-HM, a GP solver specifically designed for high-frequency and multi-scale PDEs. On a set of benchmark tasks, GP-HM shows promising performance. This might motivate alternative directions of developing machine learning solvers. In the future, we plan to develop more powerful optimization algorithms to further accelerate the convergence and to investigate GP-HM in a variety of practical applications.

## Acknowledgments

This work has been supported by MURI AFOSR grant FA9550-20-1-0358, NSF CAREER Award IIS-2046295, and and NSF OAC-2311685.

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

# Appendix

## A Covariance Function Derivation

In this section, we show how to obtain our covariance function in (8) of the main paper. We leverage the fact that the student $t$ density is a scale mixture of Gaussians with a Gamma prior over the inverse variance,

$$p(x|\mu, a, b) = \int_0^\infty \mathcal{N}(x|\mu, \tau^{-1})\text{Gam}(\tau|a, b)\text{d}\tau$$

$$= \frac{b^a}{\Gamma(a)}\left(\frac{1}{2\pi}\right)^{1/2}\left[b + \frac{(x-\mu)^2}{2}\right]^{-a-1/2}\Gamma(a+1/2). \tag{17}$$

The key to obtain this is to leverage the form of the normalizer of the Gamma distribution. When merging terms in the Gaussian and Gamma prior in the integration, one can construct another unnormalized Gamma distribution. Accordingly, the integration w.r.t $\tau$ gives rises to the normalizer.

If we set $\nu = 2a$ and $\lambda = a/b$, we immediately obtain the standard student $t$ density,

$$\text{St}(x|\mu, \lambda, \nu) = \frac{\Gamma(\nu/2 + 1/2)}{\Gamma(\nu/2)}\left(\frac{\lambda}{\pi\nu}\right)^{1/2}\left[1 + \frac{\lambda(x-\mu)^2}{\nu}\right]^{-\nu/2-1/2}, \tag{18}$$

where $\mu$ is the mean, $\lambda$ is the precision (inverse variance) parameters, and $\nu$ is the degree of freedom.

Next, we observe that the spectral density of a Matérn covariance function is a student $t$ density (Rasmussen and Williams, 2006). Given the Matérn covariance

$$\gamma_{\nu,\rho_q}(x, x') = \frac{2^{1-\nu}}{\Gamma(\nu)}\left(\sqrt{2\nu}\frac{|x-x'|}{\rho_q}\right)^\nu K_\nu(\sqrt{2\nu}\frac{|x-x'|}{\rho_q}), \tag{19}$$

the spectral density is $\text{St}(s; 0, 4\pi^2\rho^2, 2\nu)$. That means,

$$\gamma_{\nu,\rho}(\Delta) = \int_{-\infty}^\infty \text{St}(s; 0, 4\pi^2\rho^2, 2\nu)\exp\{i2\pi s \cdot \Delta\}\text{d}s, \tag{20}$$

where $\Delta = |x - x'|$. From the scale-mixture form (17), we can set $\hat{a} = \nu$ and $\hat{b} = \hat{a}/(4\pi^2\rho^2)$, and obtain

$$\text{St}(s; 0, 4\pi^2\rho^2, 2\nu) = \int_0^\infty \mathcal{N}(s|0, \tau^{-1})\text{Gam}(\tau|\hat{a}, \hat{b})\text{d}\tau. \tag{21}$$

Substituting (21) into (20), we have

$$\gamma_{\nu,\rho}(\Delta) = \int_0^\infty \text{Gam}(\tau|\hat{a}, \hat{b})\int_{-\infty}^\infty \mathcal{N}(s|0, \tau^{-1})\exp\{i2\pi s \cdot \Delta\}\text{d}s\text{d}\tau. \tag{22}$$

Consider the inverse Fourier transform,

$$\int_{-\infty}^\infty \text{St}(s; \mu, 4\pi^2\rho^2, 2\nu)\exp(i2\pi\Delta \cdot s)\text{d}s$$

$$= \int_0^\infty \text{Gam}(\tau|\hat{a}, \hat{b})\int_{-\infty}^\infty \mathcal{N}(s|\mu, \tau^{-1})\exp(i2\pi s \cdot \Delta)\,\text{d}s\text{d}\tau, \tag{23}$$

we observe that

$$\mathbb{F}^{-1}[\mathcal{N}(s|\mu, \tau^{-1})] = \int \mathcal{N}(s|\mu, \tau^{-1})\exp(i2\pi s \cdot \Delta)\,\text{d}s$$

$$= \exp(-2\pi^2\tau^{-1}\Delta^2)\exp(i2\pi\mu \cdot \Delta)$$

$$= \mathbb{F}^{-1}[\mathcal{N}(s|0, \tau^{-1})]\exp(i2\pi\mu \cdot \Delta)$$

$$= \int \mathcal{N}(s|0, \tau^{-1})\exp(i2\pi s \cdot \Delta)\,\text{d}s \cdot \exp(i2\pi\mu \cdot \Delta), \tag{24}$$

where $\mathbb{F}^{-1}$ is the inverse Fourier transform, and $i$ indicates complex numbers. Note that when we set $\mu = 0$, from the second line, we see $\mathbb{F}^{-1}[\mathcal{N}(s|0, \tau^{-1})] = \exp\left(-2\pi^2\tau^{-1}\Delta^2\right)$. That means, the inverse transform just moves out a Fourier basis with frequency $\mu$.

Substitute (24) into (22), we obtain

$$
\int_{-\infty}^{\infty} \mathrm{St}(s; \mu, 4\pi^2\rho^2, 2\nu) \exp(i2\pi\Delta \cdot s)\mathrm{d}s
$$
$$
= \int_0^{\infty} \mathrm{Gam}(\tau|\hat{a}, \hat{b}) \int_{-\infty}^{\infty} \mathcal{N}(s|0, \tau^{-1}) \exp\left(i2\pi s \cdot \Delta\right) \mathrm{d}s\mathrm{d}\tau \cdot \exp(i2\pi\mu \cdot \Delta)
$$
$$
= \gamma_{\nu,\rho}(\Delta) \cdot \exp(i2\pi\mu \cdot \Delta).
$$

Therefore, when we model the spectral density $S(s)$ as a mixture of student-t distribution,

$$
S(s) = \sum_{q=1}^{Q} w_q \left(\mathrm{St}(s; \mu_q, 4\pi^2\rho_q^2, 2\nu) + \mathrm{St}(s; -\mu_q, 4\pi^2\rho_q^2, 2\nu)\right), \tag{25}
$$

It is straightforward to obtain the following covariance function,

$$
k_{\mathrm{StM}}(x, x') = \sum_{q=1}^{Q} w_q \cdot \gamma_{\nu,\rho_q}(x, x') \cos(2\pi\mu_q(x - x')). \tag{26}
$$

## B  Experimental Settings

**The Poisson Equation**. We considered 1D and 2D Poisson equations with different source functions that lead to various scale information in the solution. We used Dirichlet boundary conditions.

$$
u_{xx} = f(x), \quad x \in [0, 2\pi],
$$
$$
u_{xx} + u_{yy} = f(x, y), \quad (x, y) \in [0, 2\pi] \times [0, 2\pi]. \tag{27}
$$

For the 1D Poisson equation, we created source functions $f$ that give the following high-frequency and multi-frequency solutions, $u_1 = \sin(100x)$, $u_2 = \sin(x) + 0.1\sin(20x) + 0.05\cos(100x)$, $u_3 = \sin(6x)\cos(100x)$, and $u_4 = x\sin(200x)$. In addition, we tested with a *challenging* hybrid solution that mixes a high-frequency with a quadratic function, $u_5 = \sin(500x) - 2(x - 0.5)^2$ where we set $x \in [0, 1]$. For the 2D Poisson equation, we tested with the following multi-scale solutions, $u_6 = \sin(100x)\sin(100y)$ and $u_7 = \sin(6x)\sin(20x) + \sin(6y)\sin(20y)$.

**Allen-Cahn Equation**. We considered 1D and 2D Allen-Cahn (nonlinear diffusion-reaction) equations with different source functions and Dirichlet boundary conditions.

$$
u_{xx} + u(u^2 - 1) = f(x), \quad x \in [0, 2\pi],
$$
$$
u_{xx} + u_{yy} + u(u^2 - 1) = f(x, y), \quad (x, y) \in [0, 1] \times [0, 1]. \tag{28}
$$

For the 1D equation, we tested with solutions $u_1 = \sin(100x)$ and $u_2 = \sin(6x)\cos(100x)$. For the 2D equation, we created the source $f$ that gives the following mixed-scale solution, $u = (\sin(x) + 0.1\sin(20x) + \cos(100x)) \cdot (\sin(y) + 0.1\sin(20y) + \cos(100y))$.

**Advection Equation.** Third, we evaluated with a 1D advection (one-way) equation,

$$
u_t + 200u_x = 0, \quad x \in [0, 2\pi], \quad t \in [0, 1]. \tag{29}
$$

We used the Dirichlet boundary conditions, and the solution has an analytical form, $u(x, t) = h(x - 200t)$ where $h(x)$ is the initial condition for which we chose as $h(x) = \sin(x)$.

**Method Implementation.** We implemented our method with JAX (Frostig et al., 2018) while all the competing ML based solvers with Pytorch (Paszke et al., 2019). For all the kernels, we initialized the length-scale to 1. For the Matérn kernel (component), we chose $\nu = 5/2$. For our method, we set the number of components $Q = 30$, and initialized each $w_q = 1/Q$. For 1D Poisson and 1D Allen-cahn equations, we varied the 1D mesh points from 400, 600 and 900. For 2D Poisson, 2D Allen-cahn and 1D advection, we varied the mesh from $200 \times 200$, $400 \times 400$ and $600 \times 600$. We chose an ending frequency $F$ from $\{20, 40, 100\}$, and initialize $u_q$'s with `linspace(0, F, Q)`.

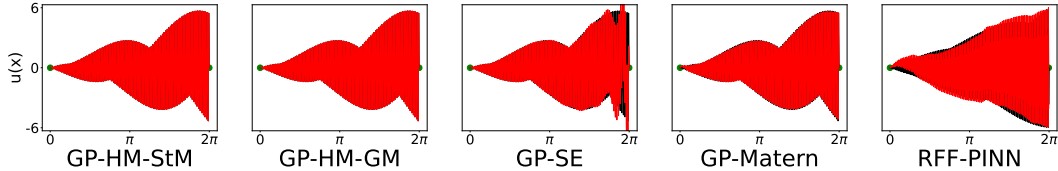

Figure 5: Prediction for the 1D Poisson equation with solution $x\sin(200x)$.

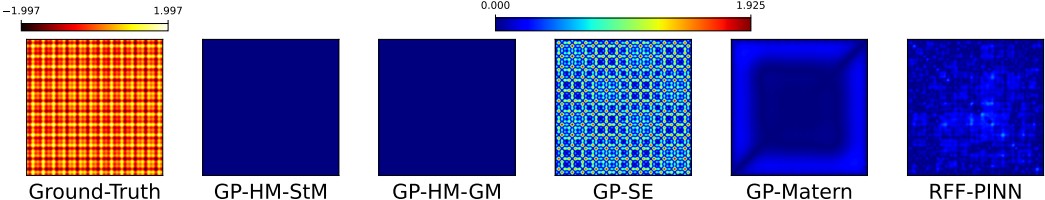

Figure 6: Point-wise solution error for 2D Poisson equation and the solution is $u(x) = \sin(6x)\sin(20x) + \sin(6y)\sin(20y)$.

We used ADAM for optimization, and the learning rate was set to $10^{-2}$. The maximum number of iterations was set to 1M, and we used the summation of the boundary loss and residual loss less than $10^{-6}$ as the stopping condition. The solution estimate $\mathcal{U}$ was initialized as zero. We set the $\lambda_b = 500$. For W-PINN, we varied the weight of the residual loss from $\{10, 10^3, 10^4\}$. For Rowdy, we combined `tanh` with $\sin$ activation, $\phi(x) = \tanh(x) + \sum_{k=2}^{K} n\sin((k-1)nx)$. We followed the original Rowdy paper (Jagtap et al., 2022) to set the scaling factor $n = 10$ and varied $K$ from 3, 5 and 9. For the spectral method, we used 200 Trigonometric bases, including $\cos(nx)$ and $\sin(nx)$ where $n = 1, 2, \ldots, 100$. We used the tensor-product for the 2D problems and 1D advection. We used the least mean square method to estimate the basis weights. To run RFF-PINNs, we need to specify the number and scales of the Gaussian variances to construct the random features. To ensure a broad coverage, we varied the number of variances from $\{1, 2, 3, 5\}$. For each number, we set the variances to be the commonly used values suggested by authors, $\{1, 20, 50, 100\}$, combined with randomly sampled ones. The detailed specification is given by Table 3. There are in total 20 settings. We report the best result of RFF-PINN from these settings. For all the PINN based methods, we varied the number of collocation points from 10K and 12K.

| Number | Scales |
|---|---|
| 1 | $1, 20, 50, 100, \mathrm{rand}(1, [1, K])$ |
| 2 | $3 \times \mathrm{rand}\left(2, \{1, 20, 50, 100, \mathrm{rand}(1, [1, K])\}\right), 2 \times \mathrm{rand}(2, [1, K])$ |
| 3 | $3 \times \mathrm{rand}\left(3, \{1, 20, 50, 100, \mathrm{rand}(1, [1, K])\}\right), 2 \times \mathrm{rand}(3, [1, K])$ |
| 5 | $2 \times \{1, 20, 50, 100, \mathrm{rand}(1, [1, K])\}, 3 \times \mathrm{rand}(5, [1, K])$ |

Table 3: The number and scales of the Gaussian variances used in RFF-PINN, where $\mathrm{rand}(k, \mathcal{A})$ means randomly selecting $k$ elements from the set $\mathcal{A}$ without replacement, $l\times$ means repeating the sampling to generate $l$ configurations, and $K$ is the maximum candidate frequency for which we set $K = 200$.

## C   More Results

### C.1   Learning Behavior and Computational Efficiency

We examined the training behavior of our method. As shown in Fig. 8, with the covariance based on the student $t$ mixture, GP-HM can converge faster or behave more robustly during the training. Overall, in most cases, GP-HM with covariance based on the student $t$ mixture performs better than with Gaussian mixture.

The computation efficiency of GP-HM is comparable to PINN-type approaches. For example, on solving 1D Poisson and Allen-cahn equations, the average per-iteration time of GP-HM (mesh 200), PINN and RFF-PINN are 0.006, 0.004 and 0.004 seconds. For 2D Poisson and Allen-cahn equations

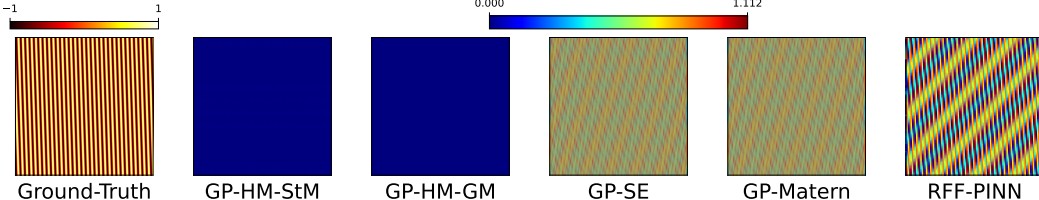

Figure 7: Point-wise solution error for 1D Advection equation and the solution is $\sin(x - 200t)$.

| *Method* | 1D | | | | | 2D | |
|---|---|---|---|---|---|---|---|
| | $u_1$ | $u_2$ | $u_3$ | $u_4$ | $u_5$ | $u_6$ | $u_7$ |
| PINN | 622 | 688 | 624 | 610 | 619 | 4,275 | 5,355 |
| RFF-PINN | 562 | 546 | 576 | 555 | 544 | 3,394 | 5,493 |
| Spectral method | 502 | 495 | 600 | 480 | 517 | 5,778 | 7,062 |
| Chebfun | 1.05 | 1.22 | 1.19 | 1.38 | 3.90 | N/A | N/A |
| Finite Difference | 1.25e-02 | 1.27e-2 | 1.22e-2 | 1.22e-2 | 1.22e-2 | N/A | N/A |
| GP-HM-GM | 536 | 1,858 | 775 | 703 | 3,510 | 4,173 | 5,561 |
| GP-HM-StM | 683 | 2,164 | 914 | 852 | 4,263 | 5,263 | 6,435 |

Table 4: Running time in seconds in solving 1D and 2D Poisson equations, where $u_j$'s are different high-frequency and multi-scale solutions: $u_1 = \sin(100x)$, $u_2 = \sin(x) + 0.1\sin(20x) + 0.05\cos(100x)$, $u_3 = \sin(6x)\cos(100x)$, $u_4 = x\sin(200x)$, $u_5 = \sin(500x) - 2(x - 0.5)^2$, $u_6 = \sin(100x)\sin(100y)$ and $u_7 = \sin(6x)\sin(20x) + \sin(6y)\sin(20y)$.

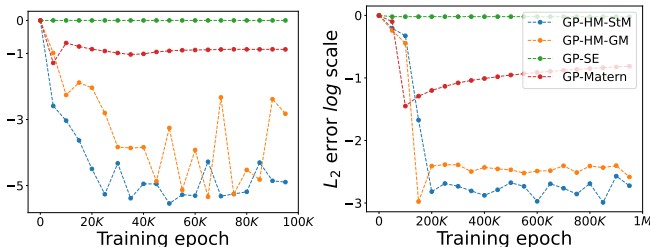

(a) 1D Poisson with solution $u_3$.   (b) 2D Poisson with solution $u_7$

Figure 8: The learning curve.

and 1D advection, the average per-iteration time of GP-HM (mesh $200 \times 200$) is 0.022 seconds while PINN and RFF-PINN (with two scales) took 0.006 and 0.02 seconds, respectively. We examined the running time on a Linux workstation with NVIDIA GeForce RTX 3090 GPU. Thanks to the usage of the grid structure and the product covariance, our GP solver can scale to a large number of collocation points, without need for additional low rank approximations.

We also reported the total running time for every test case in Table 4 and 5. We can see that the running time of GP-HM is comparable to PINN and RFF-PINN in most cases. However, the ML based solvers are slower than traditional methods. This might be because the ML solvers use optimization to find the solution approximation while the numerical methods often use interpolation and fixed point iterations, which are usually more efficient.

## C.2 Influence of Collocation Point Quantity

We examined how the number of collocation points influences the solution accuracy. To this end, we tested with a 1D Poisson and 2D Poisson equation, whose solutions include high frequencies. In Fig. 9, we show the solution accuracy with different grid sizes (resolutions). We can see that in both PDEs, using low resolutions gives much worse accuracy, e.g., less than 200 in 1D and $200 \times 200$ in 2D Poisson. The decent performance is obtained only when resolutions is high enough, e.g., 300 in 1D and $400 \times 400$ in 2D Poisson. That means, the number of collocation points is large (particularly for

| Method | 1D Allen-cahn | | 2D Allen-cahn | 1D Advection |
|---|---|---|---|---|
| | $u_1$ | $u_2$ | | |
| PINN | 509 | 828 | 2,509 | 2,496 |
| RFF-PINN | 1,227 | 1,172 | 4,421 | 2,495 |
| Spectral method | 504 | 552 | 3,840 | 2,188 |
| Chebfun | 6.57 | 6.0 | N/A | 1.39 |
| Finite Difference | 2.32e-1 | 2.36e-1 | 1,130 | 12.6 |
| GP-HM-StM | 735 | 2,291 | 7,447 | 2,574 |
| GP-HM-GM | 612 | 2,013 | 6,238 | 2,239 |

Table 5: Running time in seconds solving 1D, 2D Allen-cahn equations and 1D advection equation, where $u_1$ and $u_2$ are two test solutions for 1D Allen-cahn: $u_1 = \sin(100x)$, $u_2 = \sin(6x)\cos(100x)$. The test solution for 2D Allen-cahn is $(\sin(x) + 0.1\sin(20x) + \cos(100x)) \cdot (\sin(y) + 0.1\sin(20y) + \cos(100y))$, and for 1D advection equation is $\sin(x - 200t)$.

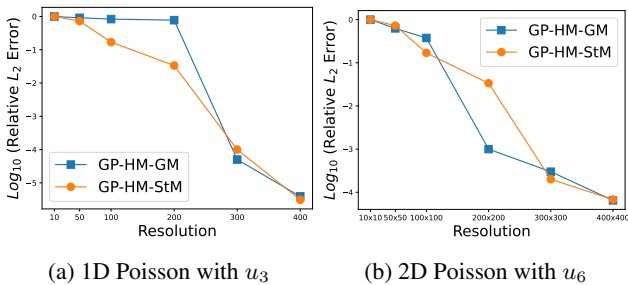

(a) 1D Poisson with $u_3$                 (b) 2D Poisson with $u_6$

Figure 9: The solution error using different grid resolutions.

2D problems, e.g., 160K collocation points for the resolution $400 \times 400$). However, it is extremely costly or practically infeasible for the existent GP solvers to incorporate massive collocation points, due to the huge covariance matrix. Our GP solver (defined on a grid) and computational method can avoid computing the full covariance matrix, and highly efficiently scale to high resolutions. The results have demonstrated the importance and value of our model and computation method.

## D   Limitation and Discussion

The learning of GP-HM can automatically prune useless frequencies and meanwhile adjusts $\mu_q$ for the preserved components, namely, those with nontrivial values of $w_q$, to align with the true frequencies in the solution. However, the selection and adjustment of the covariance components often require many iterations, like tens of thousands, see Fig. 8a. More interestingly, we found that the first-order optimization approaches, like ADAM, perform well, yet the second-order optimization, which in theory converges much faster, such as L-BFGS, performs badly. This might be because the component selection and adjustment is a challenging optimization task, and might easily encounter inferior local optimums. To overcome this limitation and challenge, we plan to try with alternative sparse prior distribution over the weights $w_q$, such as the horse-shoe prior and the spike-and-slab prior, to accelerate the pruning and frequency learning. We also plan to try other optimization strategies, such as alternating updates of the component weights and frequencies, to see if we can accelerate the convergence and if we can embed and take advantage of the second-order optimization algorithms.

