# OpenReview forum: "Solving High Frequency and Multi-Scale PDEs with Gaussian Processes"
_ICLR.cc/2024/Conference — ICLR 2024 poster_

### Official Review · Reviewer_J8Ew · 2023-10-16

**Soundness:** 3 good
**Presentation:** 2 fair
**Contribution:** 2 fair
**Rating:** 6
**Confidence:** 5

**Summary:**

The paper discusses the numerical solution of multi-scale partial differential equations (PDEs) with Gaussian processes (GPs).
To this end, the submission proposes to use variations of the spectral mixture kernel as a prior model.
Additionally, selecting a covariance function that factorises into the product of dimension-wise covariances implies a Kronecker structure in the covariance matrices.
As a result, the algorithm requires less-than-cubic complexity to evaluate the loss function, and a user can optimise the PDE solution and (hyper)parameters efficiently.
Experiments demonstrate the advantage of this method over physics-informed neural networks and those Gaussian-process-based solvers that use more traditional prior distributions.

**Strengths:**

The topic of the paper (machine-learning-based PDE solvers) is of growing interest in the ICLR community and contributes some valuable concepts to this area. More specifically, the following contributions are novel, to the best of my knowledge:

* Using a spectral mixture kernel for solving multi-scale PDEs
* Numerically optimising the joint distribution of PDE solution and PDE information can be efficient if a Kronecker-factorised prior is used (something that is not necessarily the case for more traditional MLE or MAP estimation without further assumptions).

The manuscript is easy to follow for a reader who is roughly familiar with physics-informed neural networks, PDEs, and GPs.
Overall, I consider this to be a good paper.

**Weaknesses:**

The weaknesses of this submission are mostly presentational (with two exceptions that concern the experiments).
I believe the following issues can _and should_ all be corrected before publication.


1. The manuscript misses a range of closely related work about solving partial differential equations with Gaussian processes:

    * Howhadi [1] discusses Bayesian numerical homogenisation (solving multi-scale PDEs with GPs), a central piece of related work that this paper should address prominently.
    * Accessing the derivatives of the Gaussian process as a function of $\vec{\mathcal{U}}$ (the trick suggested in Equation (13)) is discussed in Section 2 and Section 3 in the paper by Krämer et al. [2].
    * The related work section (Section 5) mentions the work on solving PDEs with GPs by Chen et al. (2021) and Long et al. (2022b) but completely misses all work that belongs to the field of probabilistic numerical algorithms (see, for instance, [1-3] below and the references therein).

    I am aware that large parts of the ICLR community may be more familiar with physics-informed neural networks than with Gaussian-process-based differential-equation solvers and that taking this into account for the exposition therefore makes sense for an ICLR paper. However, the works mentioned above are closely related and must be discussed appropriately.

2. I am surprised the submission never mentions traditional approaches to solving multi-scale problems, such as those based on numerical homogenisation (a starting point could be [4]). I know that the paper targets the ICLR community and that large parts of this community may be more interested in machine-learning-based PDE solvers than traditional PDE solvers. Still, existing approaches should be credited by (at least) mentioning their existence. (This relates to point 4. below)

3. I am also surprised that the experiments only discuss the precision of the approximations rather than the work required to achieve this precision. The experiments in Appendix C suggest that the per-iteration runtime of the proposed algorithm is comparable to that of a physics-informed neural network. However, Figure 9 indicates that tens of thousands of iterations are needed. What is each algorithm's overall runtime (training time) to achieve the precisions in Tables 1 and 2?

4. What would we find if we included a non-machine-learning-based solver in the experiments and compared the runtime to the accuracy of all these methods? I would like to see this comparison. I am not saying that the proposed method must outperform traditional solvers, which have been studied and optimised for a long time, but that the context is critical.


5. This could be personal taste, but I think the paper might benefit from a slightly more rigorous notation surrounding Equation (14), for instance, by clarifying which joint probability density Equation (14) is the logarithm of. I am imagining something like mentioning $$
\mathcal{L} = \log p(\mathcal{U}, \mathcal{H} \mid \Theta, \tau_1, \tau_2)$$ somewhere at the beginning of Section 4. Contextualising this term against the maximum-a-posteriori and maximum-likelihood loss for the same estimation problem would be even better. What do the authors think?


I consider points 1 and 2 essential and 3 and 5 slightly less important to resolve before publication.
I expect all four of those issues to be relatively straightforward to fix.
Point 4 would be optional, as incorporating it is non-trivial. However, I would like to see those results.

I am open to revising my score once these issues have been addressed.





**References:**

[1] Howhadi, Bayesian numerical homogenization. Multiscale Modeling \& Simulation. 2015.

[2] Krämer et al.: Probabilistic numerical method of lines for time-dependent partial differential equations. AISTATS. 2022.

[3] Hennig et al.: Probabilistic numerics. Cambridge University Press. 2022.

[4] Altmann et al.: Numerical homogenisation beyond scale separation. Acta Numerica. 2021.

**Questions:**

None

---

> ### Author Response · Authors · 2023-11-15
> **Thanks for your insightful and constructive comments**
>
> Thanks for your insightful and constructive comments. Here are our responses. C: comments; R: responses
>
> >C1: "The manuscript misses a range of closely related work about solving partial differential equations with Gaussian processes"
>
> R1: Thanks much for providing these excellent references. We do agree these works are related and important. We will cite and discuss them in our paper to appreciate their contributions.
>
> >C2: "I am surprised the submission never mentions traditional approaches to solving multi-scale problems, such as those based on numerical homogenisation (a starting point could be [4]). I know that the paper targets the ICLR community and that large parts of this community may be more interested in machine-learning-based PDE solvers than traditional PDE solvers. Still, existing approaches should be credited by (at least) mentioning their existence. (This relates to point 4. below)"
>
> R2: We appreciate the reviewer suggesting and providing references for traditional approaches.  We do agree. We will add references and credit those methods (including [4]) in our paper.
>
> >C3: "I am also surprised that the experiments only discuss the precision of the approximations rather than the work required to achieve this precision. The experiments in Appendix C suggest that the per-iteration runtime of the proposed algorithm is comparable to that of a physics-informed neural network. However, Figure 9 indicates that tens of thousands of iterations are needed. What is each algorithm's overall runtime (training time) to achieve the precisions in Tables 1 and 2?"
>
> R3: Great observation and questions. Actually, the number of iterations needed to reach a decent accuracy depends on the problem. That is why we report the average per-iteration time. In most cases, our method needs  <100K iterations to achieve the accuracy reported in the tables, e.g., 10K for $u_1$ and 20K for $u_3$ for 1D Poisson. In the most challenging cases, including $u_5$ for 1D Poisson and $u_7$ for 2D Poisson, it needs more iterations (>=200K). This is also comparable to the number of iteration required in PINN based methods, e.g., in [1], the PINN is trained with 10K ADAM iterations and 50K L-BFGS iterations.
> Thanks for the great suggestion. We will supplement the detailed total running time for each test case in our paper.
>
> [1] Shibo Li, Michael Penwarden, Yiming Xu, Conor Tillinghast, Akil Narayan, Robert Kirby, and Shandian Zhe, “Meta Learning of Interface Conditions for Multi-Domain Physics-Informed Neural Networks”, The 40th International Conference on Machine Learning (ICML), 2023.

---

> ### Author Response · Authors · 2023-11-15
>
> >C4: "What would we find if we included a non-machine-learning-based solver in the experiments and compared the runtime to the accuracy of all these methods? I would like to see this comparison. I am not saying that the proposed method must outperform traditional solvers, which have been studied and optimised for a long time, but that the context is critical."
>
> R4: Great suggestion! We actually have included the result of a Spectral method based on Fourier bases plus least mean square (denoted by "Spectral method") in our tables. We followed your advice to test with three other non-ML solvers. The first one is finite difference (discretion resolution = 400, no less than the number of collocation points used in our method), implemented by [PyPDE](https://py-pde.readthedocs.io/en/latest/examples_gallery/poisson_eq_1d.html#sphx-glr-examples-gallery-poisson-eq-1d-py) library. The second one is [Chebfun](https://www.chebfun.org/) that uses Chebshev polynomials. The third one is spectral Galerkin method (with legendre and chebyshev polynomials, 48 quadrature nodes), implemented by [shenfun](https://shenfun.readthedocs.io/en/latest/) library. We list the results on solving the 1D Poisson equations in the paper. We highligh our method and the best of the other methods.
>
> We can see that finite difference though gives reasonable results, its accuracy is consistently inferior to our method. The solution accuracy of Chebfun is consistently high. Spectral Galerkin, however, seems to fail in all the cases. The three methods run much faster than our method and PINN. This is reasonable, because the ML solvers use optimization to approximate the solution rather than interpolation. We will supplement the results for all the test PDEs in our paper.
>
> |  Relative $L_2$ Error  | $u_1$ |   $u_2$ |  $u_3$   | $u_4$   | $u_5$   |
> | :---        |            ---: |---:  | ---:  | ---:  | ---:
> | GP-HM-StM    | 6.53e-7   | 2.71e-3   |   3.17e-6  | 8.97e-6 | 4.22e-4|
> | GP-HM-GM  | 3.99e-7      | 2.73e-3   | 3.92e-6   |1.55e-6|1.82e-3|
> | Spectral Method (Fourier + LMS)   |   2.36e-2    | 3.47e0   | 1.02e0   |1.02e0|9.98e-1||
> | Finite Difference   |   5.58e-1    | 4.78e-2   | 2.34e-1   |1.47e0|1.40e0|
> | Chebfun   |   3.05e-11   | 1.17e-11   | 5.81e-11   |1.14e-10|8.95e-10|
> | Spectral Galerkin(legendre)   |   4.59e3    | 3.05e2   | 2.62e3   |2.72e4|1.29e3|
> | Spectral Galerkin (chebyshev)   |   9.49e2   | 7.84e2   | 1.99e3   |2.55e4|1.05e3|
>
> >C5: "This could be personal taste, but I think the paper might benefit from a slightly more rigorous notation surrounding Equation (14), for instance, by clarifying which joint probability density Equation (14) is the logarithm of. I am imagining something like mentioning "
>
> R5: This is a great suggestion. We will add the clarification of the joint probability (as suggested) in Eq4 to make the whole probabilistic modeling structure (learning objective) more clear.

---

> > ### Comment · Reviewer_J8Ew · 2023-11-20
> >
> > Thanks for the reply!
> >
> > R4: I missed the spectral method; thanks for clarifying. I appreciate the additional experiments, too. When including those results in the paper, I would like to see a mention of the total runtimes to achieve those numbers (which relates to Reviewer FJgw's comment about "chebfun").
> >
> > Overall, thanks for promising all those revisions.
> > I am open to revising my score upon seeing the promised updates in the PDF.

---

> > > ### Author Response · Authors · 2023-11-21
> > >
> > > Thanks for your feedback. We have added the results of running non-ML solvers (including chebfun) and total running time in our revision (which has been uploaded). We highlighted the new results, discussions and settings in blue in the experimental section. Please feel free to leave additional comments and suggestions. We'd love to continue improving our paper.

---

> ### Author Response · Authors · 2023-11-21
> **updated manuscript  (near the end of rebuttal)**
>
> Dear Reviewer J8Ew,
>
> We have followed your suggestions to **update the new version of our manuscript**, where  **new results, like running non-ML solvers (including chebfun), total running time, and polished statements are added and highlighted in blue**.
>
> Since the End of author/reviewer discussions is just in one day, may we know if our response addresses your main concerns?  We are happy to engage in more discussion and paper improvements.
>
> Thank you again for reviewing our paper!
>
> Authors

---

> > ### Comment · Reviewer_J8Ew · 2023-11-22
> >
> > Thanks for the updates.
> >
> > I am happy to see the included runtimes and have updated my score accordingly.
> >
> > Some revisions were promised but did not make it into the revision (e.g. discussion of numerical homogenization or probabilistic numerics, plus some presentational edits). Still, I recommend accepting this paper now.
> >
> > Thanks to the authors for the active discussion!

---

> > > ### Author Response · Authors · 2023-11-22
> > >
> > > We thank you so much for the valuable support!

---

### Official Review · Reviewer_FJgw · 2023-10-30

**Soundness:** 2 fair
**Presentation:** 3 good
**Contribution:** 2 fair
**Rating:** 5
**Confidence:** 3

**Summary:**

In this paper, the authors consider the problem of solving high-frequency partial differential equations (PDEs) using a Gaussian Process approach. The aim is to propose an alternative to PINNs, which struggle for high-frequency or multiscale PDE using a GP framework. To do so, the authors consider a prior distribution of the solution to the PDE as a mixture of student t distributions to capture high frequencies. Then, they propose a computationally fast GP method, which consists of constructing a sample grid on the domain through a Cartesian product of points, which is then exploited in the algorithm.

**Strengths:**

- The paper is well-structured and the problem is clearly introduced and motivated.
- The authors perform several numerical experiments on synthetic datasets, which show that their method outperform PINNs in terms of resulting accuracy of the solution.

**Weaknesses:**

- One of the weaknesses of the paper is that the difference between prior GP-based works in the literature is unclear. The authors should clearly state the differences between their work and the existing ones, in particular the papers by Chen et al (JCP, 2021) and Raissi et al (JCP, 2017), cited in this work. I believe that the two differences are: (1) The use of student-t distribution and (2) using a Kronecker product grid to speed-up computations. In the current version, the title and abstract of the paper might be misleading to readers in the sense that this paper is not the first one to consider solving a PDE with GP.
- I am very surprised by the Spectral method experiment performed by the authors. A quick experiment with Chebfun (https://www.chebfun.org/) allows me to solve the 1D Poisson equation to 10 digits accuracy in a fraction of a second.
- The authors state at the end of Section 5 that their work is "the first to [...] use the Kronecker product structure to efficiently solve PDEs". I believe this sentence is erroneous. It is completely natural when using spectral methods to exploit the Kronecker product structure of the domain (see e.g. Fortunato & Townsend, 2019).
- The authors offer very little conclusion and the main limitation section from the Appendix should appear in the main text. One of the key limitation of the method is that it is limited to simple geometry, where one can use a Kronecker product structure. However, this severely limits the applicability of the method and questions its advantages over spectral methods.

**Questions:**

- Is there any concrete advantage of using the Student t distribution over the Gaussian distribution? The authors state that the Student t distribution is beneficial for high frequency but the performance seems very similar in the experiments of Table 1.
- The experiments in Section 6 should provide the source terms (at least in Appendix) to allow for reproducibility.
- Equation (6) should include the domain on which the integration is performed.
- Minor comment: 2nd sentence of p.2: "the performance of this method is unstable" -> "this method is unstable"
- Section 6: Timings should be reported in main text as it's one of the main advantage of the proposed method.

---

> ### Author Response · Authors · 2023-11-15
> **Thanks for the many valuable questions and comments**
>
> Thanks for the many valuable questions and comments. Here are our clarification and responses. C: comments; R: responses
>
> > C1: "One of the weaknesses of the paper is that the difference between prior GP-based works in the literature is unclear. The authors should clearly state the differences between their work and the existing ones, in particular the papers by Chen et al (JCP, 2021) and Raissi et al (JCP, 2017), cited in this work. I believe that the two differences are: (1) The use of student-t distribution and (2) using a Kronecker product grid to speed-up computations. In the current version, the title and abstract of the paper might be misleading to readers in the sense that this paper is not the first one to consider solving a PDE with GP."
>
> R1: In fact, we did explicitly emphasize the key difference with Chen et al (JCP, 2021). Please see the bottom of page 4 and top of page 5. In Section Related Work, we have also discussed the challenge of the SE/Matern kernel used by Chen et al (JCP, 2021) in solving the high-frequency/multi-scale PDEs (4th line, 2nd paragraph), which motivates our design of mixture kernel (a 3rd critical difference).  We agree your summary about the differences as well. We believe the key difference of Raissi et al (JCP, 2017) has also been covered by our paper --- see Section Releated Work, the 1st sentence of 2nd paragraph --- it only solves **linear PDEs with noisy measurement of source terms**. Our work (as well as Chen et al (JCP, 2021)) is more general in that it applies to both linear and nonlinear PDEs. We will highlight the difference in a stronger manner.
>
> Second, we **never** claim/imply that our paper is "the first one to consider solving a PDE with GP". In fact, In Introduction Section (2nd paragraph, Page 2), we explicitly claim that **"we resort to an alternative arising ML solver framework, Gaussian processes (GP)" with (Chen et al., 2021) cited**. We will emphasize more on this point.
>
> > C2: "I am very surprised by the Spectral method experiment performed by the authors. A quick experiment with Chebfun (https://www.chebfun.org/) allows me to solve the 1D Poisson equation to 10 digits accuracy in a fraction of a second."
>
> R2: Our spectral method was implemented based on Fourier bases to be consistent with kernel construction in our method for a fair comparison. The performance is indeed inferior (as shown in the tables). This is to show that introducing a mixture model in the frequency domain (rather than the temporal domain), with which to construct the GP, is much more effective.
>
> Second, we do agree there can be more effective traditional solvers, such as Chebfun  (but not all the numerical methods work better for this problem; please see R4 to Reviewer J8Ew below). In many forward problems, machine-learning (ML) based PDE solvers, like GP or PINN, have not been able to perform equally well (in accuracy/efficiency) as traditional solvers. However, as explained in the 1st paragraph of Introduction, ML-based solvers "are simple to implement, and can solve inverse problems efficiently and conveniently". In addition, GP models can easily, robustly handle noisy measurements and/or source terms, offering a principled and convenient framework to quantify and reason under uncertainty. We therefore view **developing ML-based solvers as a promising direction, like the concurrent blooming efforts in this direction, even in several aspects the ML-based solvers are now still behind traditional numerical solvers that have been developed for decades**.
>
> > C3: "The authors state at the end of Section 5 that their work is "the first to [...] use the Kronecker product structure to efficiently solve PDEs". I believe this sentence is erroneous. It is completely natural when using spectral methods to exploit the Kronecker product structure of the domain (see e.g. Fortunato & Townsend, 2019)."
>
> R3: Thanks for the comment. We do agree that our statement is not precise. We mean to say GP plus Kronecker product structure (rather than Kronekcer product structure only). We will revise our statement to be accurate and clear. Please see R2 to Reviewer GK3t above for the detailed clarification.

---

> ### Author Response · Authors · 2023-11-15
>
> > C4: "The authors offer very little conclusion and the main limitation section from the Appendix should appear in the main text. One of the key limitation of the method is that it is limited to simple geometry, where one can use a Kronecker product structure. However, this severely limits the applicability of the method and questions its advantages over spectral methods."
>
> R4: Due to the space limit, we leave the limitation section in Appendix. We will re-organize the paper to merge the limitation and conclusion section, and put it in the main paper.
>
> We believe our method is **NOT** limited to simple geometry. Even if the domain is not a grid, we can create a grid that covers the domain, and use the GP conditional mean (see Eq4 and Eq13) to predict the boundary values and fit the boundary conditions via the GP likelihood $\text{log} \mathcal{N}(\mathbf{g}|\mathbf{u}_b, \tau_1^{-1}\mathbf{I})$ (see Eq14). We then maximize the log joint probability to solve the problem. Note that the computation efficiency (via Kronecker product plus tensor algebra) remains the same.
>
> > C5: "Is there any concrete advantage of using the Student t distribution over the Gaussian distribution?..."
>
> R5: Great question. As explained in Section 3 (see the text under Eq5), the student $t$ distribution has a fat-tailed density, and hence can more robustly capture "the (potentially many) minor frequencies". By contrast, the Gaussian distribution density has a very thin tail, and can be much more sensitive to minor frequencies, especially when they are not ignorable.
>
> However, we agree that our most test cases comprise only one or just a few principle frequencies, and hence the results do not exhibit a strong contrast of the two choices. But two examples worth attention. One is the 1D Poisson with solution $u_4 = x\cdot sin(200 x)$, and the other is 1D Poisson with solution $u_5 = sin(500x) - 2(x-0.5)^2$. In both cases, the solution contains many minor frequencies (due to non-trigonometric terms). We can see that our method with student $t$ mixture (GP-HM-StM) achieves much smaller error (one magnitude smaller) than with Gaussian mixture (GP-HM-GM).
>
> We will add more such test cases to highlight their difference.
>
> > C6: "The experiments in Section 6 should provide the source terms (at least in Appendix) to allow for reproducibility."
>
> R6: Thanks for the suggestion. As explained in Section 6, specifically, the text under Eq17, Eq18 and Eq19, we follow existing literature to first craft the solution, e.g., $u_1 = \sin(100x)$ and $u_2=\sin(x) + 0.1\sin(20x) + 0.05\cos(100x)$, and then apply the differentiation operator to obtain the source term $f$. We will supplement differential results (e.g., $f$) in the paper.
>
> > C7: "Section 6: Timings should be reported in main text as it's one of the main advantage of the proposed method."
>
> R7: Thanks for the suggestion. We will add the time table to the main paper.

---

> > ### Comment · Reviewer_FJgw · 2023-11-20
> >
> > I read the authors' reply and thank them for their detailed response. However, I am still not fully convinced by the pertinence of the paper's contributions.
> >
> > The authors acknowledge that ML-based solvers `` have not been able to perform equally well (in accuracy/efficiency) as traditional solvers'', and mention that ML solvers are simple to implement and handle noisy measurements. I am not suggesting that the authors should necessarily outperform standard numerical solvers but at least show challenging experiments (as they are the main contributions of the paper) that display a promising competitive advantage.
> > I do not think that claiming that the present approach outperforms existing techniques on high frequency data is a fair claim given that standard approaches (such as finite element method or spectral methods) come with rigorous error bounds and convergence rates to the solution as the resolution increases, which the authors' approach does not have due to the complex optimization process. Hence, the authors first reported a high error using spectral methods for one experiment but updated it to 10^{-10} error in the response to Reviewer J8Ew.
> >
> > Concerning the complexity of the geometry, the authors reply that one can always embed the domain into a square with a uniform grid. I agree with this comment but the number of points required to approximate the domain geometry would likely grow extremely fast, which would result in a much less efficient representation than using a meshing algorithm combined with a finite element solver.
> >
> > Given the above concerns about the relevance and potential applicability of the paper's contributions, I would like to keep my current score.

---

> ### Author Response · Authors · 2023-11-21
>
> > "Hence, the authors first reported a high error using spectral methods for one experiment but updated it to 10^{-10} error in the response to Reviewer J8Ew."
>
> We believe this has already been clarified in our paper and response R2.
>
> First, spectral method contains a large number of instances. Chebfun is just one instance, which adopts Chebshev polynomials as bases. One shouldn't view Chebfun is the only choice of spectral methods. You can have many other choices. In fact, different bases can end up with different performance. In our paper, we used the Fourier bases (the most straightforward choice), and it does show inferior performance. As explained in R2, we believe putting such a baseline is important to highlight the advantage of designing a mixture model in the frequency space.  As reported in R4 to J8Ew, we have also tried spectral Galerkin methods, which completely failed in our ``simple'' cases.
>
> Second, we would love to come out more test cases and more challenging cases. But even for the current test cases, finite difference --- which is another commonly used and fundamental numerical approaches --- have already been surpassed by our methods. Though Chebfun works very well in 1D cases, it **cannot solve** 2D stationary problem, like 2D Poisson and 2D Allen-Cahn in our paper; please see the updated draft. Similarly, can you write a Chebfun solver to solve a 2D Darcy-flow? The usage of Chebfun is limited, even in the set of cases tested in our paper.
>
> Third, circling back to the points we have made for ML based solvers. One biggest advantage of ML based solvers we believe is that they are really convenient to implement and use --- **they are much more general in that you don't need to change the modeling and computational framework to handle different PDEs.** After 70+ years' development, traditional methods contain many useful tools/tricks. If you pick up the right tool/trick, it indeed works well; but how to pick up and combine various tools for different PDEs to make them work is always a headache. Of course, ML solvers. which start to get attention for only a few years, are far from mature, in both practice and theory. We therefore view such immaturity as a sign of new promising direction, like many researchers do, rather an excuse to suppress the direction.
>
> > "Concerning the complexity of the geometry, the authors reply that one can always embed the domain into a square with a uniform grid. I agree with this comment but the number of points required to approximate the domain geometry would likely grow extremely fast, which would result in a much less efficient representation than using a meshing algorithm combined with a finite element solver. "
>
> We would like to point out that, even if the statement is true (though rigorous proof is unkown), including a large number of points on the boundary will not be an issue in that various stochastic mini-batch optimization can efficiently handle those points, just like training neural networks (or PINNs) with a huge number of data points. There is not any memory or computational bottleneck. Beside, it is very easy to leverage the modern distributed ML platform, e.g., [1],  to further speed-up, which is much cheaper than HPC/super computers.
>
> [1] Li, M., Andersen, D. G., Park, J. W., Smola, A. J., Ahmed, A., Josifovski, V., ... & Su, B. Y. (2014). Scaling distributed machine learning with the parameter server. In 11th USENIX Symposium on operating systems design and implementation (OSDI 14) (pp. 583-598).

---

### Official Review · Reviewer_SL7y · 2023-10-31

**Soundness:** 2 fair
**Presentation:** 2 fair
**Contribution:** 2 fair
**Rating:** 6
**Confidence:** 2

**Summary:**

The authors propose a solver for partial differential equations (PDE), specifically designed for high-fidelity and multi-scale PDEs, using Gaussian Process. Current PDE solver methods can be highly unstable and sensitive to hyperparameters. The authors bypass this problem by modeling the solution in the frequency domain and estimate target frequencies from the covariance function of the Gaussian process. The authors also propose an efficient algorithm to scale up the learning algorithm.

**Strengths:**

The authors described their method in detail, analyzed the runtime complexity, and provided multiple sets of experiments.

**Weaknesses:**

The paper abstract right now is very hard to follow for interested readers, instead of focussing on what the authors did in their methodologies, it should focus mainly on the contributions from a high level. The introduction clarifies the authors’ motivation well, however certain rearrangement of figures can help make it better. When the authors say their method is focussed on high fidelity and multi-scale PDEs, they can exhibit this with a figure (perhaps by moving figure 2,3 up in the first 2 pages).

“While effective, the performance of this method is unstable, and is highly sensitive to the number and
scales of the Gaussian variances, which are difficult to choose beforehand” - this statement should be associated with an example figure/experiment/prior work.

**Questions:**

See weakness section.

---

> ### Author Response · Authors · 2023-11-15
> **Thanks for your valuable and constructive comments**
>
> Thanks for your valuable comments and suggestions. We will polish our paper and improve the presentation to highlight the contributions in a more concise and easy-to-understand way, especially for readers who are not experts in Gaussian process, kernel methods and PDE solving.  We will also follow your suggestions to move figure 2 and 3 up to better articulate the problem we aim to solve, and add additional analysis and examples to illustrate the learning behaviors of random Fourier PINNs.

---

### Official Review · Reviewer_GK3t · 2023-10-31

**Soundness:** 3 good
**Presentation:** 3 good
**Contribution:** 3 good
**Rating:** 6
**Confidence:** 4

**Summary:**

This work uses Gaussian processes to solve partial differential equations. The authors propose to use a spectral mixture kernel and learn the mixture weights from data with a sparsity-inducing prior. To achieve scalability, they place collocation points on a grid and assume a product kernel which induces Kronecker structure in the resulting covariance matrices. The approach is then evaluated on three common model PDEs in one and two dimensions.

**Strengths:**

Overall, the paper presents a promising idea for how to solve certain classes of PDEs with Gaussian processes. The idea of using a sparsity-inducing prior over the frequencies of a spectral mixture kernel seems very useful and the paper demonstrates the robustness and applicability of the approach.

**Weaknesses:**

There are some weaknesses in the paper regarding the claimed novelty of the approach, the experimental evaluation and the proper attribution of existing approaches.

## Originality
In my opinion the following two statements from the paper are incorrect or at least too strong for what is presented in the paper:

1. Quote from the Abstract (also mentioned in contributions): "The covariance derived from the Gaussian mixture spectrum corresponds to the known spectral mixture kernel. We are the ﬁrst to discover its rationale and effectiveness for PDE solving." In my opinion, this is incorrect. Härkönen et al. (2023) construct a specific kernel for linear PDEs, which as they write explicitly at the end of Section 4.1 recovers the spectral mixture kernel (Wilson et al. 2013) as a special case.
2. "By contrast, for PDE solving, it is natural to estimate the solution values on a grid, which opens the possibility of using Kronecker products for efﬁcient computation. To our knowledge, our work is the ﬁrst to realize this beneﬁt and use the Kronecker product structure to efﬁciently solve PDEs."I disagree with this being the first instance of Kronecker product structure being used to efficiently solved PDEs. First of all, Kronecker product structure (on regular grids) is being used to solve PDEs (either via preconditioners or via the connection of separation of variables and tensor numerical methods (e.g. Gavrilyuk et al. 2019)). Second, as the authors mention themselves in the manuscript, the computational efficiency of product kernels on regular grids is well-known for GPs (Saatci, 2012). Further, PDE solvers based on GPs which use tensor product kernels (specifically Matern) have also been proposed previously (e.g. Wang et al. 2021).

## Quality of Presentation
The paper is generally well-written and easy to understand. The ideas were presented in a coherent way and accessible to readers outside the specific subfield. The plots were largely informative, but I would have preferred larger plots to make them more legible.

## Theoretical Results
The theoretical results are mostly standard applications of known results and look correct to me, although I did not check every derivation in detail.

## Experimental Evaluation
The experimental evaluation is largely well-done. I have some questions about the baselines that are compared against. Specifically, the vanilla GP baseline with a Matern kernel. Is this using a product kernel? The authors write in the paper that "[...] it is extremely costly or practically infeasible for the existent GP solvers to incorporate massive collocation points, due to the huge covariance matrix.". Why should one not be able to leverage the Kronecker product + Toeplitz structure (see Sections 3.1 and 3.2 of Wilson et al. 2015) of a product of Matern kernels for the vanilla GP case as well? If this was not done here, the baseline that is presented here seems rather weak. I can imagine that a scale mixture still outperforms a vanilla Matern kernel, but the performance gap in terms of time and memory should be significantly smaller I would expect.

## Related Work
The paper discusses related work from the domain of PINNs well, but misses some of the work on Gaussian-process based PDE solvers. For example, the following papers used GPs to solve PDEs:

- J. Cockayne, C. Oates, T. Sullivan, M. Girolami, *Probabilistic numerical methods for PDE-constrained Bayesian inverse problems*, AIP Conf. Proc. 1853 (2017)
- Wang, Junyang, et al. *Bayesian numerical methods for nonlinear partial differential equations.* Statistics and Computing 31, 2021 URL: https://arxiv.org/abs/2104.12587
- Pförtner, Marvin, et al. *Physics-informed Gaussian process regression generalizes linear PDE solvers.* arXiv preprint arXiv:2212.12474 (2022).
- Chen, Yifan, Houman Owhadi, and Florian Schäfer. *Sparse Cholesky factorization for solving nonlinear PDEs via Gaussian processes.* arXiv preprint arXiv:2304.01294 (2023).

**References**
Papers cited in the review for reference:

- Härkönen, Marc, Markus Lange-Hegermann, and Bogdan Raita. *Gaussian Process Priors for Systems of Linear Partial Differential Equations with Constant Coefficients.* International Conference on Machine Learning (ICML), 2023, URL: https://arxiv.org/abs/2212.14319
- Wilson, Andrew, and Ryan Adams. *Gaussian process kernels for pattern discovery and extrapolation.* International Conference on Machine Learning (ICML), 2013, URL: https://arxiv.org/abs/1302.4245
- Wang, Junyang, et al. *Bayesian numerical methods for nonlinear partial differential equations.* Statistics and Computing 31, 2021 URL: https://arxiv.org/abs/2104.12587
- Gavrilyuk, Ivan and Khoromskij, Boris N.. *Tensor Numerical Methods: Actual Theory and Recent Applications* Computational Methods in Applied Mathematics, vol. 19, no. 1, 2019, pp. 1-4. URL: https://doi.org/10.1515/cmam-2018-0014
- Wilson, Andrew Gordon, Christoph Dann, and Hannes Nickisch. “Thoughts on Massively Scalable Gaussian Processes.” arXiv, November 5, 2015. URL: https://doi.org/10.48550/arXiv.1511.01870.

**Questions:**

- What about the marginal uncertainty output by the Gaussian process? Does it capture the approximation error of the mean or is it miscalibrated?
- What's the impact of the number of components of the mixture kernel on the approximated solution? An ablation such as a plot of number of components vs error would be informative. Also how does it affect the difficulty of the optimization problem? I could imagine choosing too many components makes the optimization problem significantly harder to solve in practice.
- Do the kernel matrices you define have Toeplitz structure? Stationary kernels on a 1D grid would have. This could further accelerate the required computations (see Section 3.2. of Wilson et al. 2015).

---

> ### Author Response · Authors · 2023-11-15
> **Thanks for your detailed and constructive comments**
>
> We thank the reviewer for their careful and constructive comments.
>
> >C1: "... Härkönen et al. (2023) construct a specific kernel for linear PDEs, which as they write explicitly at the end of Section 4.1 recovers the spectral mixture kernel as a special case"
>
> R1: Thanks for the excellent reference. We will modify our claim. Actually, our work is done roughly at the same time; this is not the first submission (and we did not release our work on ArXiv). Anyway, we will remove the statement, cite and discuss this nice work.
>
> >C2: "...I disagree with this being the first instance of Kronecker product structure being used to efficiently solved PDEs..."
>
> R2: Thanks for the valuable comments.  Our statement is indeed **not** precise, and we will remove the claim. From the context, we mean to say, we are using GP + Kronecker product structure --- **taking advantage of the corresponding computational advantage** --- to solve PDEs (rather the Kronecker product itself). The computational advantage is fulfilled by combining the properties of Kronecker product of kernel matrices and tensor algebra, which we believe is rarely used in the domain of PDE solving. In fact, we have acknowledged tensor-product structures are widely used in PDE solving; see the text under Eq11.
>
> We never claimed that our work invents/discovers the computation advantage of GP+Kronecker product. As emphasized in related work, this advantage has been known in the GP community; our contribution is to **enable this computational advantage** in PDE solving.
>
> We do agree (Wang et al. 2021) have also proposed the product kernel, but the paper is mainly about theoretical analysis (e.g., the sample path property) and does not fulfill efficient computation. Thanks this nice paper --- it provides additional strong justification of using a product kernel; we will cite and discuss about it. Thus we believe, the contributions of our work are still orthogonal to existing work in a large degree. We will make a more detailed, careful discussion and clarification.
>
> >C3: "...the vanilla GP baseline with a Matern kernel. Is this using a product kernel?..."
>
> R3: Great question. We did use product kernels for vanilla GP baselines. We use the same computational approach (Kronecker product + tensor algebra) to fulfill efficient training and prediction. We will clarify it in the paper.
>
> >C4: More reference on GP based PDE solvers
>
> R4: We appreciate the reviewer for providing these great references. We will cite and discuss them in our work.
>
> >C5: "What about the marginal uncertainty output by the Gaussian process?..."
>
> R5: Great question! We show the result on 1D Poisson (with solution $u_1$ and $u_3$) in [Fig. 1](https://github.com/wctghlbgtat/Solving-High-Frequency-and-Multi-Scale-PDEs-with-Gaussian-Processes/blob/main/u1.png) and [Fig. 2](https://github.com/wctghlbgtat/Solving-High-Frequency-and-Multi-Scale-PDEs-with-Gaussian-Processes/blob/main/u3.png). We compared our method with student-t mixture (GP-HM-StM) and GP with Matern kernel (GP-matern), in terms of log variance (left), log absolute error (middle), and prediction (right). We can see that the marginal uncertainty indeed reflects the error. In both figures, GP-Matern exhibits much larger errors in almost all the locations, and its corresponding variance is much larger than GP-HM-StM as well. In addition, both GP-Matern and GP-HM-StM show smaller variance on the boundary, and meanwhile their boundary errors are very small. It reflects both methods fit the boundary well. We will supplement the result and discussion in our paper.
>
> > C6: "What's the impact of the number of components of the mixture kernel on the approximated solution? ..."
>
> R6: Great question! We have followed your suggestion to give the result on 1D Poisson (with solution $u_1$ and $u_3$). We varied the component number from {3, 10, 30, 50, 80}. See [Here](https://github.com/wctghlbgtat/Solving-High-Frequency-and-Multi-Scale-PDEs-with-Gaussian-Processes/blob/main/components.png). As we can see, using two few or too many components can both increase optimization challenges (too few can make it harder to move to the right frequency). The best choice is in between. We will supplement the results and discussion in our paper.
>
> >C7: "Do the kernel matrices you define have Toeplitz structure? Stationary kernels on a 1D grid would have. This could further accelerate the required computations (see Section 3.2. of Wilson et al. 2015)."
>
> R7: Great question. Our method does not require the grid must be evenly-space in each dimension.  In fact, we can randomly sample locations in each dimension and then generate the grid (see text above Eq12). Hence the kernel matrices do not have the Toeplitz structure in general. However, if we specify a regular (evenly-spaced) grid, the Toeplitz structure will show up and we can certainly leverage it to further accelerate the computation. Thanks for suggesting this great idea, and we will add it into our discussion.

---

> > ### Author Response · Authors · 2023-11-15
> >
> > >C8: "The plots were largely informative, but I would have preferred larger plots to make them more legible. "
> >
> > R8: Great suggestion. We will enlarge the figures to improve the legibility.

---

> > > ### Comment · Reviewer_GK3t · 2023-11-20
> > >
> > > I appreciate the author's responsiveness to include feedback from the review. I would like to see the changes to the main text included in the revised draft even if it exceeds the 9 page limit (ideally highlighted in color). Otherwise it is challenging to give a revised opinion based on the promised changes alone.
> > >
> > > > R5: Great question! We show the result on 1D Poisson (with solution and ) in Fig. 1 and Fig. 2. We compared our method with student-t mixture (GP-HM-StM) and GP with Matern kernel (GP-matern), in terms of log variance (left), log absolute error (middle), and prediction (right). We can see that the marginal uncertainty indeed reflects the error. In both figures, GP-Matern exhibits much larger errors in almost all the locations, and its corresponding variance is much larger than GP-HM-StM as well. In addition, both GP-Matern and GP-HM-StM show smaller variance on the boundary, and meanwhile their boundary errors are very small. It reflects both methods fit the boundary well. We will supplement the result and discussion in our paper.
> > >
> > > Thank you for providing this plot. The scale of the error seems to be correct. However, I would recommend plotting either squared error and variance or standard deviation and absolute error. Otherwise it's an apples to oranges comparison that makes it unnecessarily difficult for the reader. It is somewhat unfortunate that none of the error structure is captured in the uncertainty quantification (e.g. the oscillatory behavior of the error). For my own curiousity only: Do you have any intuition why that is?

---

> > > > ### Author Response · Authors · 2023-11-21
> > > >
> > > > Thanks for your feedback.
> > > >
> > > > > "I would like to see the changes to the main text included in the revised draft even if it exceeds the 9 page limit (ideally highlighted in color). Otherwise it is challenging to give a revised opinion based on the promised changes alone."
> > > >
> > > > We've uploaded a revision draft, which includes the changes to the main text as you suggested. We've highlighted the modification part with blue. We've erased the inprecise/inaccurate statement and add more discussions about the related work. Please feel free to leave further comments and suggestions so that we can keep improving our manuscript.
> > > >
> > > > > " I would recommend plotting either squared error and variance or standard deviation and absolute error. Otherwise it's an apples to oranges comparison that makes it unnecessarily difficult for the reader"
> > > >
> > > > That's a great suggestion. We will re-draw the plot and then move them into the manuscript.
> > > >
> > > > > "It is somewhat unfortunate that none of the error structure is captured in the uncertainty quantification (e.g. the oscillatory behavior of the error). For my own curiousity only: Do you have any intuition why that is?"
> > > >
> > > > Yes, we did observe this as well. We believe that it might be too optimistic to assume that the GP UQ can quantify the fine-grained error structure in practice. Our intuition is twofold. First, from the computational perspective, the predictive variance is obtained by Laplace approximation, which comes from Hessian matrix, which might be over smooth and average out local oscillation. Second, from the modeling perspective, the variance of a GP model actually does not involve its prediction or training outputs. That is, the uncertainty does not directly relate to its prediction. This might weaken the association between the prediction error and uncertainty. This is actually a well-known problem of GP models [1], and one potential alternative can be the student-t process.
> > > >
> > > > [1] Shah, Amar, Andrew Wilson, and Zoubin Ghahramani. "Student-t processes as alternatives to Gaussian processes." Artificial intelligence and statistics. PMLR, 2014.

---

> > > > ### Author Response · Authors · 2023-11-21
> > > > **updated manuscript (near the end of rebuttal)**
> > > >
> > > > Dear Reviewer GK3t,
> > > >
> > > > We have followed your suggestions to update the new version of our manuscript, where new results, like running non-ML solvers (including chebfun), total running time, and polished statements are added and highlighted in blue.
> > > >
> > > > Since the End of author/reviewer discussions is just in one day, may we know if our response addresses your main concerns? We are happy to engage in more discussion and paper improvements.
> > > >
> > > > Thank you again for reviewing our paper!
> > > >
> > > > Authors

---

> > > > > ### Comment · Reviewer_GK3t · 2023-11-22
> > > > >
> > > > > The changes made to the manuscript improve the paper. Nonetheless, I feel my score is still appropriate. The reasons are my comments about the originality of the work (which the authors confirmed). Further, I share some of the concerns about the experimental evaluatiom raised by the other reviewers.

---

> > > > > > ### Author Response · Authors · 2023-11-22
> > > > > >
> > > > > > We thank you so much for your valuable review and suggestion!

---

### Author Response · Authors · 2023-11-20

We appreciate all reviewers' time and efforts in evaluating our work! In view of the limited available time, we would kindly like to ask the reviewers to please engage in a discussion with us (if not) given the submitted rebuttals so we can respond to the possible new questions in time.

---

### Meta-Review · Area_Chair_F2Pn · 2023-12-06

**Metareview:**

This work improves the runtime of probabilistic PDE solvers by proposing a new spectrally sparse kernel with learned spectrum. Using a Kronecker set of collocation points then allows efficient computations.

All reviewers pointed out that the framework used in the paper is not new, and that the paper is occasionally a bit too confident in claiming novelty. I urge the authors to include the numerous new references to existing work suggested by the reviewers. But they also agree that the work does contribute new techniques that improve runtime significantly. Reviewer `FJgw` also asks for more context relative to classic PDE solvers. I agree with the authors that, since probabilistic numerical methods are still evolving rapidly, they do not necessarily need to demonstrate dominance over classic methods all the time. In fact, it is precisely through the kind of small and technically intricate but crucial improvements like the ones demonstrated in this paper that the original classical methods came to achieve their present strength.

I thus recommend accepting this paper.

**Justification For Why Not Higher Score:**

I share the reviewers' worry that the paper is too confident about its own contributions, so it should perhaps not be elevated further.

**Justification For Why Not Lower Score:**

It is, however, technically intricate, sound and valuable, and worthy of presentation.

---

### Decision · Program_Chairs · 2024-01-16

Accept (poster)